# De novo macrocyclic peptides dissect energy coupling of a heterodimeric ABC transporter by multimode allosteric inhibition

Erich Stefan[1†], Richard Obexer[2†], Susanne Hofmann[1], Khanh Vu Huu[3], Yichao Huang[2], Nina Morgner[3], Hiroaki Suga[2*], Robert Tampé[1*]

[1]Institute of Biochemistry, Biocenter, Goethe University Frankfurt, Frankfurt, Germany; [2]Department of Chemistry, Graduate School of Science, The University of Tokyo, Tokyo, Japan; [3]Institute of Physical and Theoretical Chemistry, Goethe University Frankfurt, Frankfurt, Germany

**Abstract** ATP-binding cassette (ABC) transporters constitute the largest family of primary active transporters involved in a multitude of physiological processes and human diseases. Despite considerable efforts, it remains unclear how ABC transporters harness the chemical energy of ATP to drive substrate transport across cell membranes. Here, by random nonstandard peptide integrated discovery (RaPID), we leveraged combinatorial macrocyclic peptides that target a heterodimeric ABC transport complex and explore fundamental principles of the substrate translocation cycle. High-affinity peptidic macrocycles bind conformationally selective and display potent multimode inhibitory effects. The macrocycles block the transporter either before or after unidirectional substrate export along a single conformational switch induced by ATP binding. Our study reveals mechanistic principles of ATP binding, conformational switching, and energy transduction for substrate transport of ABC export systems. We highlight the potential of de novo macrocycles as effective inhibitors for membrane proteins implicated in multidrug resistance, providing avenues for the next generation of pharmaceuticals.

**\*For correspondence:**
hsuga@chem.s.u-tokyo.ac.jp (HS);
tampe@em.uni-frankfurt.de (RT)

[†]These authors contributed equally to this work

**Competing interests:** The authors declare that no competing interests exist.

## Introduction

ATP-binding cassette (ABC) transporters utilize the energy of ATP binding and hydrolysis to move a vast variety of chemically distinct compounds across cell membranes (*Rees et al., 2009*; *Thomas and Tampé, 2020*; *Locher, 2016*; *Szakács et al., 2014*). They are implicated in many physiological processes and diseases including cystic fibrosis (*Stutts et al., 1995*; *Csanády et al., 2019*), diabetes (*Aguilar-Bryan and Bryan, 1999*), lipid-trafficking disorders (*Brooks-Wilson et al., 1999*), antibiotic resistance (*Orelle et al., 2019*), and acquired drug resistance in cancer chemotherapy (*Kartner et al., 1983*; *Litman et al., 2000*). ABC transporters are composed of two transmembrane domains (TMDs), forming the substrate translocation pathway, and two nucleotide-binding domains (NBDs), converting the chemical energy of ATP binding and hydrolysis into mechanic transitions (*Rees et al., 2009*; *Locher, 2016*; *Thomas and Tampé, 2018*; *Oldham et al., 2008*). The heterodimeric multidrug resistance transporter TmrAB from *Thermus thermophilus* shares structural and functional homology with the transporter associated with antigen processing (TAP) and can restore antigen presentation in human TAP-deficient cells (*Kim et al., 2015*; *Nöll et al., 2017*; *Zutz et al., 2011*). TmrAB is particularly suitable for mechanistic and structural studies, as the various conformers along the translocation trajectory are only accessible at elevated temperatures, thus providing active

control over the system (*Nöll et al., 2017*; *Zutz et al., 2011*; *Hofmann et al., 2019*; *Stefan et al., 2020*; *Barth et al., 2018*; *Barth et al., 2020*; *Diederichs and Tampé, 2021*).

To correlate ATP binding and hydrolysis to substrate transport, several catalytic models for ABC systems have been depicted including alternating sites (*Senior et al., 1995*), constant contact (*Jones and George, 2009*), and processive clamp/ATP switch (*Higgins and Linton, 2004*; *Janas et al., 2003*; *Abele and Tampé, 2004*). In all models, substrate translocation is coupled with conformational transitions, which are linked to the ATP hydrolysis cycle (*Thomas and Tampé, 2020*; *Szöllősi et al., 2018*). The conformational landscape of a heterodimeric ABC transporter has recently been elucidated through nine high-resolution X-ray and cryo-EM structures (*Nöll et al., 2017*; *Hofmann et al., 2019*). To functionally integrate these conformers into the substrate translocation trajectory, we performed single-turnover transport studies using a variant with a slow-down in ATP hydrolysis. Using single liposome-based translocation assays, the unidirectional substrate export along a single conformational switch induced by ATP binding was demonstrated (*Stefan et al., 2020*).

Many bioactive toxins, antibiotics, and inhibitors are macrocyclic peptides (CPs) with unnatural amino acid modifications (*Nicolaou et al., 1999*; *Obexer et al., 2017*). CPs are regarded as next-generation pharmaceutical compounds to target 'undruggable' proteins implicated in disease and diagnosis (*Morrison, 2018*; *Vinogradov et al., 2019*; *Zorzi et al., 2017*). To this end, various engineered and modified CPs are clinically approved or under clinical trials, for example, Peginesatide, Linaclotide, and Pasireotide. Compared to conventional small-molecule drugs, CPs usually display an elevated selectivity as well as an increased potency, and they are easy to manufacture (*Craik et al., 2013*; *Driggers et al., 2008*). The random nonstandard peptide integrated discovery (RaPID) system comprises the identification of synthetic/de novo CPs against desired target protein complexes to enable structural and functional investigations (*Murakami et al., 2006a*; *Murakami et al., 2006b*). As a major advantage, unnatural amino acids such as N-chloroacetyl-D-tyrosine (*Goto et al., 2008*), N-methyl alanine (*Yamagishi et al., 2011*), and carboranyl alanine (*Yin et al., 2019*) can be incorporated by the Flexible In-vitro Translation (FIT) system, relying on aminoacylating ribozymes called Flexizymes (*Goto et al., 2011*) to construct highly diverse peptide libraries. Using the RaPID system, CPs were identified for receptor tyrosine kinases (*Yin et al., 2019*), ubiquitin ligase (*Yamagishi et al., 2011*), the Zika virus protease (*Nitsche et al., 2019*), prolyl hydroxylase (*McAllister et al., 2018*), and phosphoglycerate mutases (*Yu et al., 2017*).

To dissect the catalytic cycle and energy coupling of substrate translocation of a wild-type ABC transporter, we selected and identified high-affinity macrocyclic peptides against TmrAB applying the mRNA display-based RaPID system. Enriched CPs specifically bind to TmrAB and are high-affinity allosteric inhibitors, which prevent substrate transport and ATP turnover. Two mechanistically distinct CP inhibitor classes were elucidated that arrest the ABC transporter either before or right after ATP-binding induced substrate translocation. Thus, these CPs arrest TmrAB in the inward-facing (IF) and outward-facing conformation, respectively. Our work illustrates the important aspects of ATP binding and conformational switching for productive substrate translocation by a heterodimeric ABC transporter, and it also highlights the potential of CPs targeting transient conformers of membrane protein complexes to develop next-generation drug products.

## Results

### Identification of macrocyclic peptides

We elicited macrocyclic peptide binders against a heterodimeric ABC transport complex employing the RaPID approach. mRNA-encoded peptide libraries with 10–15 randomized amino acid positions were constructed by transcription of degenerate DNA templates, ligation of a puromycin linker, followed by FIT that relies on a reconstituted *Escherichia coli* translation system. The final library size was on the order of $10^{12}$ macrocyclic peptides. Head-to-side chain cyclization was mediated through ribosomal incorporation of N-chloroacetyl-D-tyrosine at the initiator position, which post-translationally undergoes intramolecular thioether formation with a downstream cysteine (*Figure 1A*). Catalytically reduced TmrA$^{E523Q}$B (TmrA$^{EQ}$B) complexes reconstituted in lipid nanodiscs with biotinylated membrane scaffold proteins (MSP) were purified by gel filtration and used as bait on streptavidin magnetic beads for affinity selection (*Figure 1—figure supplement 1*). Empty nanodiscs were

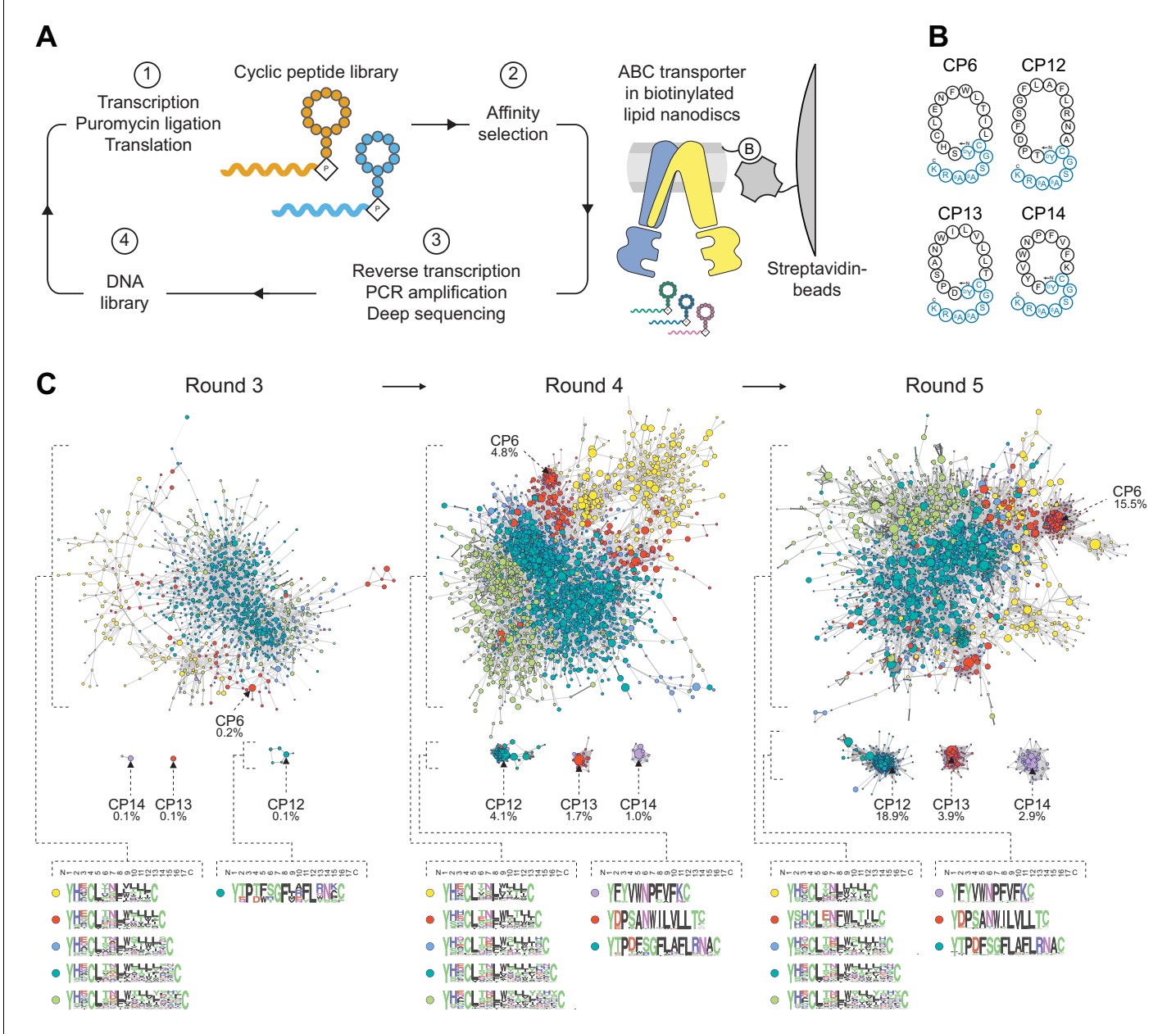

**Figure 1.** Selection of macrocyclic peptides (CPs) by random nonstandard peptide integrated discovery (RaPID). (**A**) RaPID selection of CPs. Starting from a DNA library, macrocyclic peptides were generated through transcription and ribosomal translation using the Flexible In-vitro Translation (FIT) system. Cognate mRNA was covalently attached to the nascent peptide through incorporation of mRNA-linked puromycin. ABC transporter TmrA$^{EQ}$B reconstituted in lipid nanodiscs (Nds) was immobilized on streptavidin matrices and used as bait during affinity selections. After several iterative rounds of selection, high-affinity binders were isolated and identified by deep sequencing. (**B**) CPs targeting TmrAB. Enriched CPs were produced by solid-phase synthesis and conjugated with a short linear extension (colored in blue). Head-to-side chain thioether cyclization was mediated between the N-terminal N-chloroacetyl-D-tyrosine and a cysteine residue. The C-terminal lysine was used for site-directed labeling of fluorescein ($^F$) or biotin ($^B$). (**C**) Enrichment of CPs targeting TmrAB. The 5000 most abundant macrocyclic peptides per selection round based on deep sequencing were utilized to generate sequence similarity networks (**Gerlt et al., 2015**). Nodes represent unique peptide sequences, node sizes depict peptide frequency, and node colors exhibit peptide length. Sequence alignments of peptide clusters were generated using WebLogo (**Crooks et al., 2004**).

The online version of this article includes the following source data and figure supplement(s) for figure 1:

**Source data 1.** Sequencing Data for Section of CPs.
**Figure supplement 1.** Selection and synthesis of CPs.

utilized for library clearing, purging non-target specific binders from the peptide library. After five iterative rounds of selection, including in vitro translation, affinity panning, reverse transcription, and PCR recovery, high-affinity binders were identified by deep sequencing. Along the affinity selection, several unrelated peptide families were enriched which differed in amino acid composition and length (*Figure 1B, C*). The most abundant macrocyclic peptides after the fifth selection round, CP6, CP12, CP13, and CP14, constituted 41% of the final sequenced pool and could not be grouped into families (*Figure 1C*). For further analyses, CPs were produced by solid-phase peptide synthesis and extended by a linear C-terminal tail harboring a lysine residue for site-directed labeling by fluorescein ([F]) or biotin ([B]), yielding CP[F]s and CP[B]s, respectively (*Figure 1B*). All CPs were cyclized in solution through thioether formation. In the case of CP6, the second cysteine at position 14 was utilized for cyclization, rather than cysteine at position 4, which emerged in the randomized region. CPs were purified by reversed-phase (RP) HPLC, and their correct masses were confirmed by mass spectrometry (*Figure 1—figure supplement 1*).

## Macrocyclic peptides bind with high affinity and selectivity

In order to analyze the binding affinity and specificity of CPs to TmrAB, we established a fluorescence polarization assay. Fluorescein-labeled CP[F]s were incubated with increasing concentrations of TmrAB reconstituted in liposomes at a protein-to-lipid ratio of 1:20 (w/w). The equilibrium binding revealed nanomolar binding affinities for all CP[F]s, with $K_D$ values ranging from 20 to 50 nM (*Figure 2A, B*). If empty liposomes (without reconstituted TmrAB) were added to CP[F]s, no binding was detected, demonstrating a specific interaction of all CPs with TmrAB (*Figure 2A, B*). Notably, TmrAB complexes could be affinity-captured on streptavidin beads pre-loaded with biotinylated CP[B]s and detected in the eluate but not in the flow-through (*Figure 2C*, *Figure 2—figure supplement 1*). If streptavidin beads were not pre-loaded with CP[B]s, TmrAB did not bind and was exclusively found in the flow-through. To explore the specificity of binding, we used CP13[B] and CP14[B] immobilized on streptavidin matrices to isolate intact TmrAB complexes from solubilized *E. coli* membranes. These results demonstrate the high specificity of the interaction and illustrate that the CPs can be used for efficient affinity purification of the membrane protein complex (*Figure 2D*, *Figure 2—figure supplement 1*). For all CP[F]s, kinetically stable CP[F]-TmrAB complexes were formed as confirmed by gel filtration (*Figure 2—figure supplement 1*). To address whether the CPs recognize a linear or non-linear epitope, we examined the binding of CP[F]s after disrupting the quaternary and tertiary structure of TmrAB complexes by adding anionic detergent SDS. Under these conditions, binding of TmrAB to all CP[F]s was impaired, indicating a conformation-specific interaction (*Figure 2—figure supplement 1*). We also analyzed the CP[F]-TmrAB complexes by native mass spectrometry (MS), demonstrating a stoichiometric interaction (*Figure 2E*, *Figure 2—figure supplement 2*).

## Macrocyclic peptides are potent inhibitors of ATP hydrolysis and substrate transport

In order to investigate the functional impact of CPs, we examined the effect of the CPs on ATP hydrolysis and substrate transport of TmrAB (*Figure 3A*). We reconstituted TmrAB in liposomes and first assayed the ATPase activity. In the presence of the CP[F]s, ATPase activity was completely blocked to the level of autohydrolysis (*Figure 3B*). We next set out to investigate peptide transport by TmrAB using a filter-based transport assay. To investigate whether CPs are recognized as substrates by TmrAB, we added fluorescein-labeled CP[F]s and examined ATP-dependent translocation. In contrast to the linear substrate peptide C4F, none of the CP[F]s were transported by TmrAB (*Figure 3C*). Next, we examined ATP-dependent transport of C4F peptides in the presence and absence of the different CPs. The data reveal that CP6[F] and CP12[F] completely blocked peptide translocation, while CP13[F] or CP14[F] only caused a partial inhibition (*Figure 3D*).

To elucidate the mechanistic basis for the inhibition of ATP hydrolysis and substrate transport, we investigated the effects of CPs on substrate binding. In the first approach, fluorescent CP[F]s were used as reporters in fluorescence polarization. At competitive concentrations above their $K_D$ values (*Stefan et al., 2020*), the transported peptides R9LQK or C4[ATTO655] did not affect binding of CP[F]s to TmrAB (*Figure 4A*). In the second inverse approach using CP[B]s as competitors, we did not observe a competition of C4F peptide binding, in contrast to the linear peptide C4[ATTO655], which interfered with C4F peptide binding (*Figure 4B*). In addition, we examined the impact of CP[B]s on

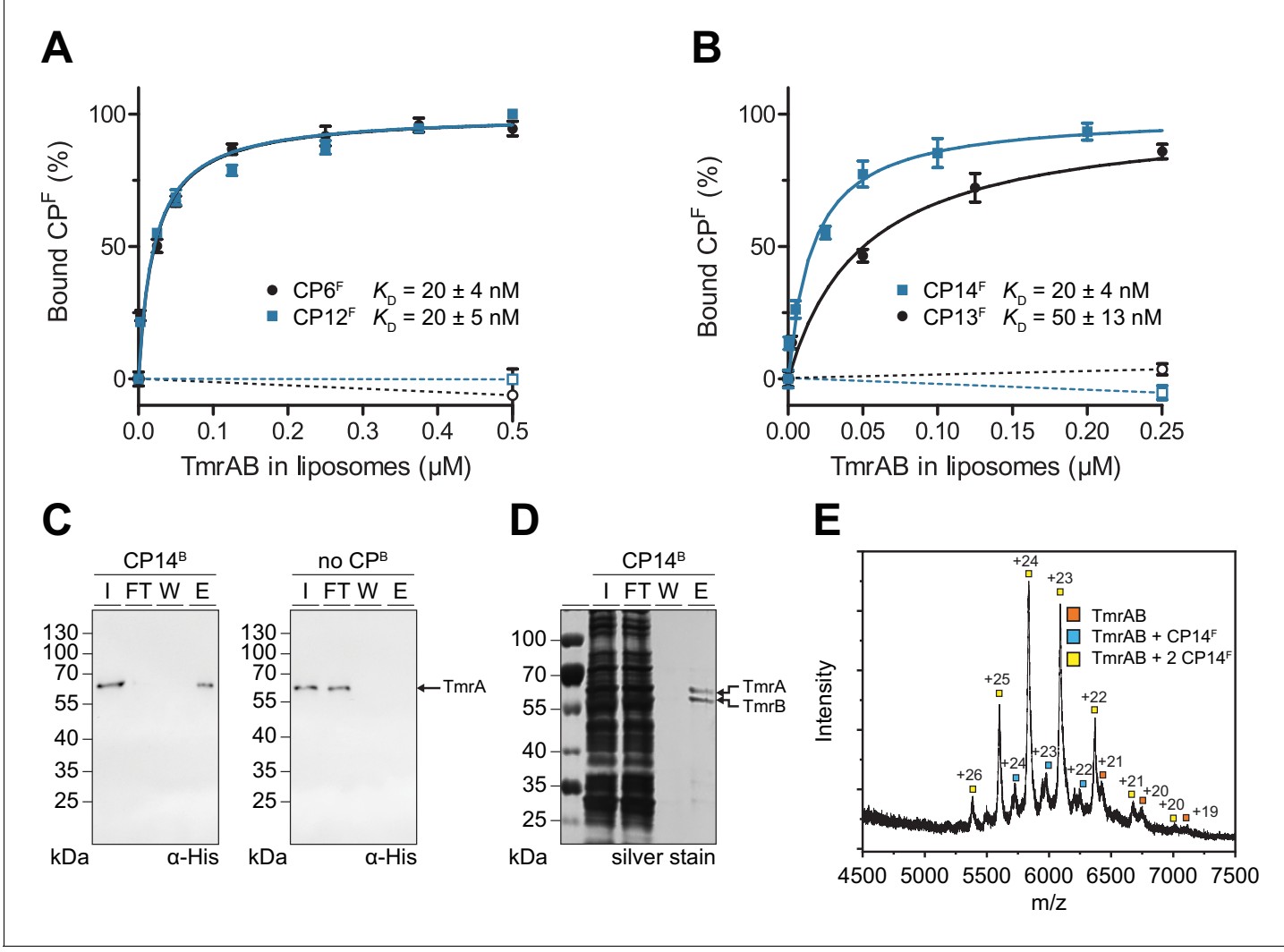

**Figure 2.** CP$^F$s specifically interact with TmrAB displaying nanomolar binding affinities. (**A, B**) Equilibrium binding analysis. Fluorescence anisotropy of CP$^F$s (50 nM) was determined at $\lambda_{ex/em}$ = 485/520 nm with increasing concentrations of TmrAB which was reconstituted in liposomes. TmrAB concentrations were calculated based on the random (50/50%) orientation in proteoliposomes (**Stefan et al., 2020**). As a control, equal amounts of empty liposomes were added to CP$^F$s (dashed lines, open symbols). The difference in fluorescence polarization was normalized to free (0%) and 100% bound CP$^F$. Mean values ± SD (n = 3) are shown, and data were analyzed by a one-site binding model. (**C**) TmrAB binds specifically to CP14$^B$-loaded matrices. Streptavidin beads loaded with CP14$^B$ (1 μM) were mixed with purified TmrAB (60 nM) for 1 hr at 4°C. Beads were washed and bound TmrAB was eluted by adding SDS loading buffer for 10 min at 95°C. Amounts of TmrA in input (I), flow-through (FT), wash fraction (W), and eluate (E) were analyzed by SDS-PAGE and immunoblotting (α-His). Beads without CP$^B$ served as negative control (right panel). (**D**) One-step purification of TmrAB via immobilized CP14$^B$ matrices. Streptavidin-agarose beads were loaded with CP14$^B$ (1 μM) and mixed with DDM-solubilized membranes of *E. coli* containing TmrAB as described in (**C**). Bound TmrAB was eluted in SDS loading buffer and analyzed by SDS-PAGE (silver stain). (**E**) Native mass spectrometry. TmrAB (4 μM) was buffer exchanged to ESI buffer and incubated with a 2-fold molar excess of CP14$^F$ for 10 min on ice. Protein complexes were investigated by ESI-TOF-mass spectrometry. Derived masses, TmrAB: 134.9 kDa, TmrAB +CP14$^F$: 137.4 kDa, TmrAB +2 CP14$^F$:140.0 kDa.

The online version of this article includes the following source data and figure supplement(s) for figure 2:

**Source data 1.** Source data for *Figure 2*.

**Figure supplement 1.** CPs specifically bind to TmrAB.

**Figure supplement 1—source data 1.** Source Data for *Figure 2—figure supplement 1*.

**Figure supplement 2.** Analysis of CP$^F$-TmrAB complexes by native mass spectrometry.

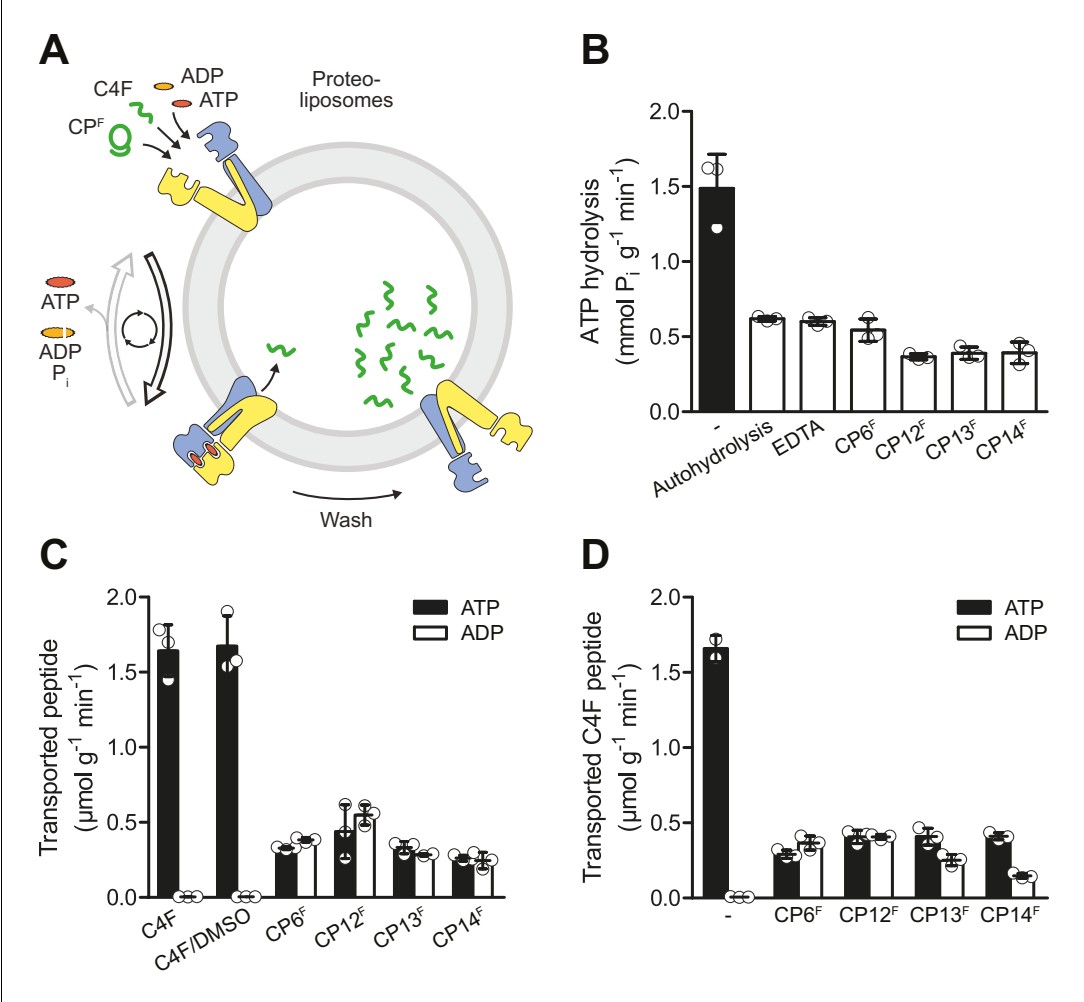

**Figure 3.** CP[F]s are potent inhibitors of ATP turnover and substrate transport. (**A**) Experimental scheme of ATP hydrolysis and peptide transport by TmrAB. (**B**) CP[F]s inhibit ATP hydrolysis. TmrAB in liposomes (100 nM) was incubated with ATP (2 mM, traced with [$\gamma^{32}$P]-ATP), and MgCl$_2$ (5 mM) in the presence and absence of CP[F]s (1 µM) for 10 min at 45°C. Autohydrolysis was determined in the absence of TmrAB, and background turnover was conducted in the presence of EDTA (10 mM). Release of [$^{32}$P] was quantified by thin layer chromatography. (**C**) CP[F]s are not transported by TmrAB. TmrAB (0.4 µM) reconstituted in liposomes was incubated with C4F peptide (3 µM) or CP[F]s (1 µM) in the presence of ATP/ADP (3 mM) and MgCl$_2$ (5 mM) for 5 min at 45°C. Since CP[F]s were dissolved in maximal 0.5% (v/v) of DMSO, C4F plus 0.5% DMSO served as control. Proteoliposomes were washed on filter plates, and transported peptides were quantified at $\lambda_{ex/em}$ = 485/520 nm. (**D**) Inhibition of substrate translocation. TmrAB in liposomes (0.4 µM) were incubated with C4F peptide (3 µM), ATP/ADP (3 mM), and MgCl$_2$ (5 mM) in the presence and absence of CP[F]s (1 µM) for 15 min at 45°C. Transported peptides were quantified as described in (**C**). In (**B–C**), mean values ± SD (n = 3) are shown.

The online version of this article includes the following source data for figure 3:

**Source data 1.** Source data for *Figure 3*.

ATP binding to TmrAB by scintillation proximity assays (SPA) (*Figure 4C*). CP12[B] and CP14[B] showed no effect on ATP binding. For CP6[B] and CP13[B], only minor effects on ATP binding were detected (*Figure 4C*). Taken together, CPs are high-affinity allosteric inhibitors of TmrAB that leave ATP binding and substrate interaction largely unaffected, while impairing ATP hydrolysis and substrate translocation across the membrane.

## Macrocyclic peptides are conformation-specific inhibitors

ABC transport complexes such as TmrAB run through multiple conformational states along the ATP hydrolysis cycle to drive unidirectional substrate transport (*Hofmann et al., 2019*; *Stefan et al., 2020*). Within the substrate translocation cycle, two major conformers, inward-facing (IF) and outward-facing (OF), were identified. To investigate whether CPs favor one of these conformers, we

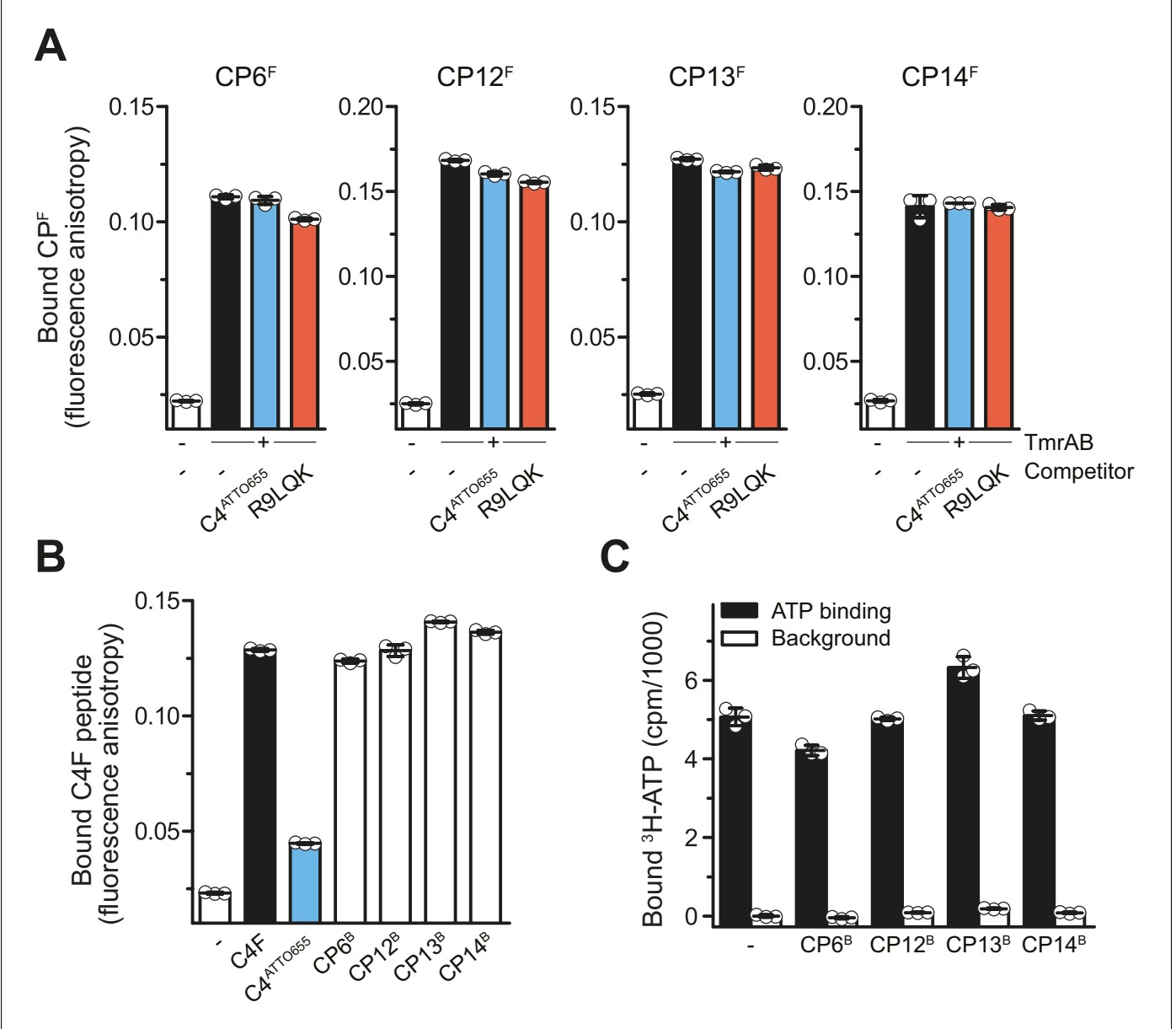

**Figure 4.** CPs do not affect peptide or ATP binding of TmrAB. (**A**) CP$^F$s binding TmrAB is not affected by substrate peptide binding. After the addition of TmrAB (0.6 µM for CP6$^F$ and CP13$^F$, 0.7 µM for CP12$^F$, 0.5 µM for CP14$^F$), fluorescence anisotropy of CP$^F$s (50 nM) were monitored at $\lambda_{ex/em}$ = 485/520 nm. For competition, C4$^{ATTO655}$ peptide (2 µM) or R9LQK peptide (200 µM) were added. (**B**) Binding of C4F peptide is not affected by CP$^B$s. C4F peptides (50 nM) were mixed with TmrAB (4 µM), and fluorescence anisotropy was monitored as described in (**A**). For competition, C4$^{ATTO655}$ (10 µM) or CP$^B$s (6 µM) were added. (**C**) CP$^B$s do not largely affect ATP binding. TmrAB (0.2 µM) were immobilized on SPA beads and incubated with ATP (3 µM, traced with $^3$H-ATP) in the presence and absence of CP$^B$s (1 µM each) for 30 min on ice. ATP binding was monitored by SPA. Background values were determined after releasing TmrAB complexes from the beads by adding imidazole (200 mM). The background signal in the absence of CP$^B$s was set to 0 cpm. In the case of CP13, the value of bound ATP exceeds the control without CPs. As this is an equilibrium experiment, this can be rationalized by the fact that CP13 stabilizes the ATP-bound OF conformer.

The online version of this article includes the following source data for figure 4:

**Source data 1.** Source data for *Figure 4*.

probed the binding of CP$^F$s by fluorescence polarization analysis. For these studies, we utilized the catalytically reduced TmrA$^{EQ}$B variant, which can be populated either in the IF (in the absence of ATP) or OF (ATP-trapped) conformation as demonstrated previously (*Stefan et al., 2020*). In the latter case, the interaction studies were performed immediately after the IF-to-OF switch by incubation

with ATP for 5 min at 45℃. To exclude an OF-to-IF return ($\tau_{1/2}$ of 25 min) (*Stefan et al., 2020*), the fluorescence anisotropy was recorded only within the first minute of incubation. Binding of CP6[F] and CP12[F] showed lower dissociation constants for the OF conformation in contrast to CP13[F] and

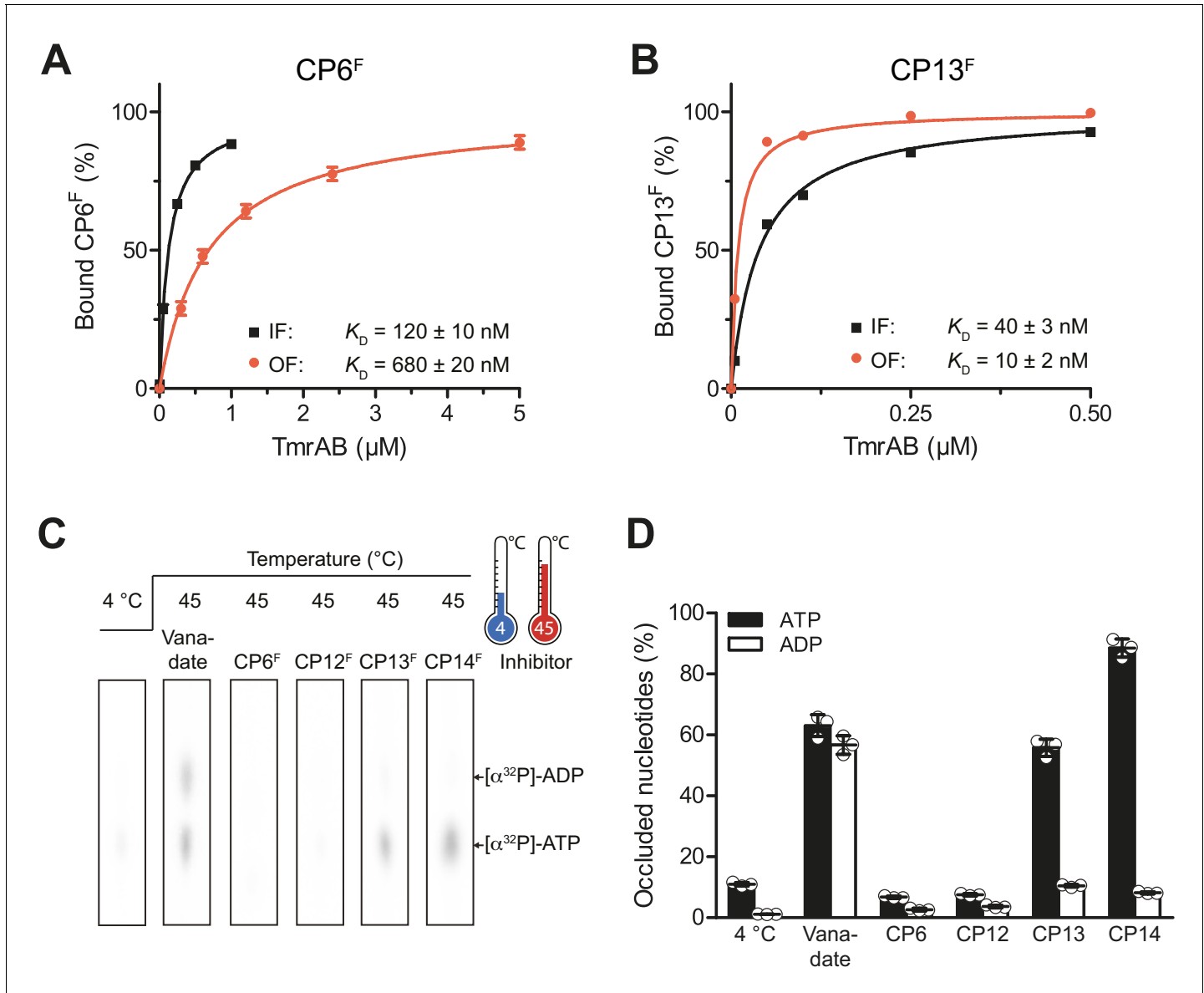

**Figure 5.** CPs bind preferentially to IF- und OF conformation and stabilize nucleotide occlusion. Conformation-specific preference of CPs. (A, B) CP6[F] (A) or CP13[F] (B, 50 nM each) were incubated with increasing concentrations of detergent-solubilized inward-facing TmrAB (in the absence of Mg-ATP) or outward-facing TmrA[EQ]B, which was trapped with Mg-ATP (1 mM) for 5 min at 45℃ as described (*Hofmann et al., 2019*; *Stefan et al., 2020*). Immediately after, the fluorescence anisotropy was assayed at $\lambda_{ex/em}$ = 485/520 nm. The difference in fluorescence polarization was normalized to free and fully bound CP[F]. Data were fitted by a one-site binding model. (C) Nucleotide occlusion promoted by CPs. TmrAB (2 µM each) were mixed with CP[F]s (4 µM), ATP (1 mM, traced with [$\alpha^{32}$P]-ATP), and MgCl$_2$ (5 mM) for 5 min at 4℃ or 45℃. Cold ATP (10 mM) was added, and freely exchangeable nucleotides were removed by rapid gel filtration. Occluded nucleotides were analyzed by thin layer chromatography and autoradiography. Representative radiograms for independent triplicates are shown. (D) Stably occluded nucleotides, [$\alpha^{32}$P]-ATP and [$\alpha^{32}$P]-ADP, were quantified by autoradiography. Data were normalized to the vanadate-trapped state. In (A, B, D), mean values ± SD (n = 3) are shown.

The online version of this article includes the following source data and figure supplement(s) for figure 5:

**Source data 1.** Source Data for *Figure 5*.
**Figure supplement 1.** Conformational arrest of TmrAB by CP[F]s.
**Figure supplement 1—source data 1.** Source Data for *Figure 5—figure supplement 1*.

CP14$^F$, which show a preference for the OF conformer (*Figure 5A, B*, *Figure 5—figure supplement 1A*). These findings were further corroborated by demonstrating that CP13$^F$ and CP14$^F$ bind in the absence and presence of ATP and ADP, while CP6$^F$ and CP12$^F$ preferentially bind under conditions where IF TmrAB is exclusively populated (*Figure 5—figure supplement 1B*).

We also examined the impact of CP$^F$s on the IF-to-OF transition of wild-type TmrAB induced by ATP binding through nucleotide occlusion (stable trapping) as the readout. We induced the IF-to-OF switch with radiotraced [α$^{32}$P]-ATP for 5 min at 45°C in the presence and absence of CPs and removed unbound nucleotides by rapid gel filtration. Subsequently, the occluded nucleotides were identified and quantified by thin layer chromatography and autoradiography (*Figure 5C, D*). As a reference, wild-type TmrAB was trapped by orthovanadate in a post-hydrolysis transition state, characterized by the stoichiometric (1:1) occlusion of ATP and ADP, which is in agreement with the 2.8 Å cryo-EM structure (*Hofmann et al., 2019*). It is important to note that nucleotide occlusion strictly requires elevated temperatures to allow a IF-to-OF conformation switch, and does not occur at 4°C (*Figure 5C, D*). In the presence of CP6$^F$ and CP12$^F$, nucleotide occlusion was completely abolished. Notably, CP13$^F$ and CP14$^F$ promoted stable ATP occlusion in wild-type TmrAB – blocking hydrolysis to ADP, which is indicative of stabilization of a pre-hydrolysis state (*Figure 5C, D*). Similar results were obtained for the catalytically reduced variant TmrA$^{EQ}$B (*Figure 5—figure supplement 1C*). In this case, CP6$^F$ and CP12$^F$ inhibited ATP occlusion and blocked the IF-to-OF transition induced by ATP binding. Conversely, as TmrA$^{EQ}$B is trapped in a pre-hydrolysis state, ATP is occluded in the presence or absence of CP13$^F$ or CP14$^F$. In conclusion, CP6$^F$ and CP12$^F$ display preferential binding to the IF conformation and inhibit the IF-to-OF switch by ATP binding. In contrast, CP13$^F$ and CP14$^F$ favorably bind to the OF conformation, promoting ATP occlusion, thus stabilizing TmrAB in a pre-hydrolysis state.

## Stabilization of a pre-hydrolysis state after substrate translocation

To investigate whether an IF-to-OF transition and conformational arrest by CP13$^F$ or CP14$^F$ can still result in a productive substrate translocation, we followed peptide transport by wild-type TmrAB by single liposome-based flow cytometry (*Figure 6A*). To this end, wild-type TmrAB was reconstituted in liposomes with a diameter of ~160 nm at a protein-to-lipid ratio of 1:20 (w/w), yielding a random (50/50) orientation, and a reconstitution efficiency of 95% as reported (*Stefan et al., 2020*). This procedure resulted in ~30 transport-competent (NBDs outside) TmrAB complexes per liposome. In flow cytometry, single liposomes were gated based on side and forward scatter intensities and then used for the evaluation of mean fluorescence intensities of transported peptide substrates (*Figure 6—figure supplement 1*). By calibration and linear regression ($R^2$ = 0.99), the mean fluorescence values of C4$^{ATTO655}$ were converted into the number of transported peptides per liposome (*Figure 6—figure supplement 1*). In the presence and absence of CP$^F$s, TmrAB-containing liposomes were incubated with C4$^{ATTO655}$ peptide and ATP for 5 min at 45°C. Along a single IF-to-OF switch and arrest by CP13$^F$ or CP14$^F$, approx. 10 or 30 C4$^{ATTO655}$ peptides per liposome were transported, respectively (*Figure 6B*). From that, a coupling ratio of 0.3 to 1.0 peptides per transporter along the ATP-induced IF-to-OF switch can be derived. These results are in agreement with the observation that CP13$^F$ or CP14$^F$ promote ATP occlusion and inhibit ATP hydrolysis of wild-type TmrAB by stabilizing a pre-hydrolysis state. In contrast, CP6$^F$ and CP12$^F$ completely prevented C4$^{ATTO655}$ peptide translocation. These findings are consistent with the results that CP6$^F$ and CP12$^F$ block C4F multiple-turnover transport (*Figure 3D*) and impair the IF-to-OF transition, as demonstrated by their preferential binding to the IF conformation and inhibition of nucleotide occlusion (*Figure 5A, B*). Taking into account that the CPs are membrane impermeable, these results indicate that the binding epitope of all CPs is on the TmrAB regions facing the cytosol.

To investigate whether CP13$^F$ and CP14$^F$ prevent multiple rounds of conformational switching after long-term incubation, C4$^{ATTO655}$ translocation by wild-type TmrAB in the presence of CP$^F$s was examined over a period of 40 min. Compared to the positive control in the absence of CP$^F$s, CP13$^F$ and CP14$^F$ drastically reduced the peptide transport rate (*Figure 6C, D*). For CP13$^F$, a single turnover driving peptide transport was observed before wild-type TmrAB became conformationally arrested (*Figure 6C*). In the presence of CP14$^F$, we noted a slow turnover of wild-type TmrAB driving peptide transport over a period of 40 min (*Figure 6C, D*). To confirm the conformational arrest of wild-type TmrAB by CP13$^F$ and the slow turnover in the presence of CP14$^F$, we examined two successive cycles of conformational transitions by flow cytometry. Liposomes containing wild-type

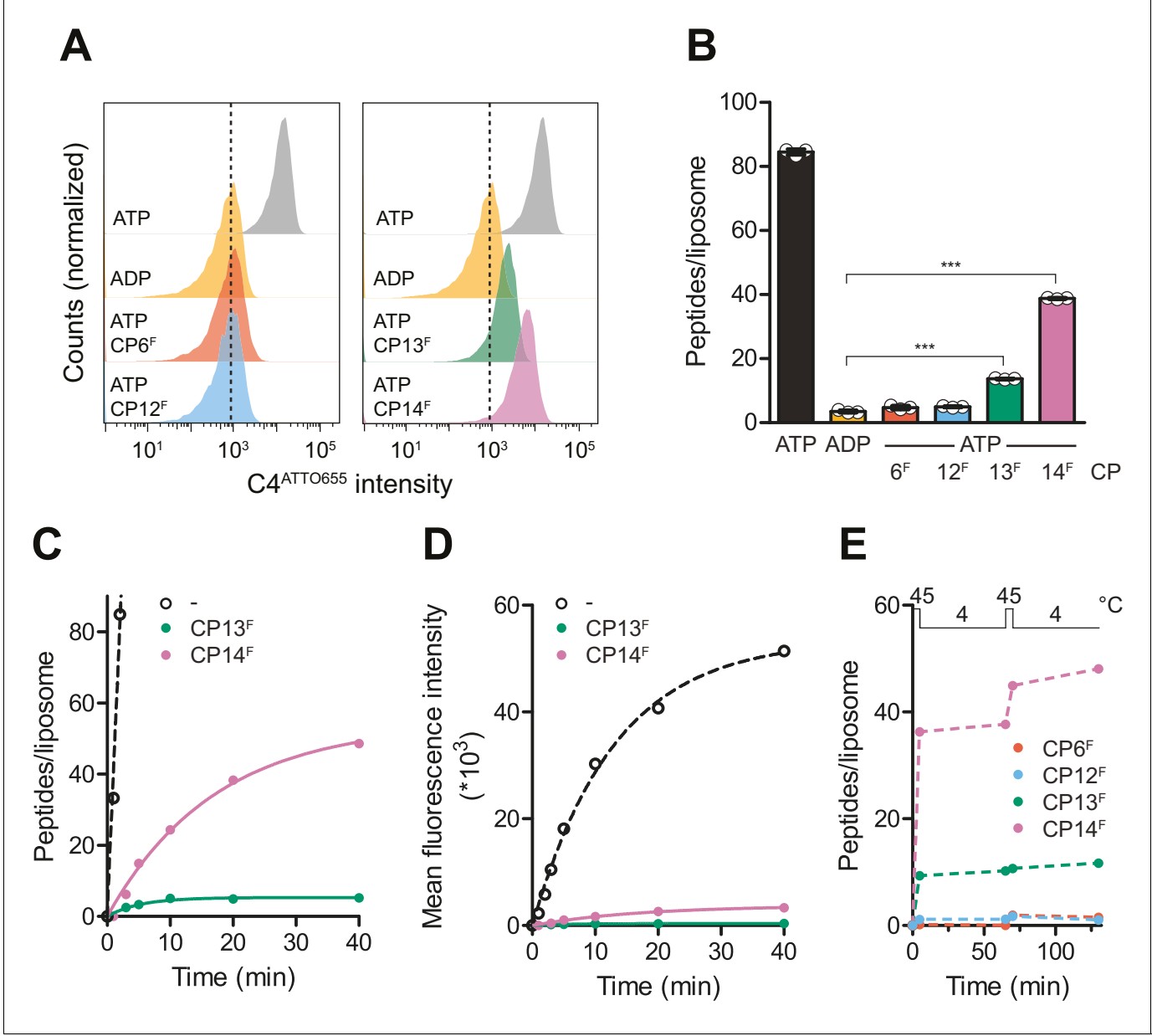

**Figure 6.** CPs block multiple-turnover transport monitored by quantitative flow cytometry. (A) Single turnover by IF-to-OF switch monitored by single-liposome flow cytometry. TmrAB in liposomes (0.4 μM) were incubated with C4$^{ATTO655}$ peptide (1 μM), ATP/ADP (3 mM), MgCl$_2$ (5 mM), and CP$^F$s (1 μM) for 5 min at 45℃. Transport reactions were stopped by the addition of EDTA (10 mM). 100,000 proteoliposomes were analyzed by flow cytometry monitoring fluorescein and ATTO655 intensities. (B) Mean fluorescence intensities of C4$^{ATTO655}$ were converted into the number of peptides per liposome using the regression analysis described above (two-tailed T-test, ***p<0.0001). (C, D) Slowdown of multiple-turnover substrate transport. TmrAB reconstituted in liposomes (0.4 μM) was incubated with C4$^{ATTO655}$ peptide (1 μM), ATP/ADP (3 mM), MgCl$_2$ (5 mM), and CP$^F$s (1 μM) for various periods of time at 45℃. Transported peptides per liposomes were evaluated and corrected by ADP background levels as described in (B). Transport kinetics were fitted monoexponentially. In (C), mean fluorescence intensities of transported C4$^{ATTO655}$ were converted into the number of peptides per liposome as described above. (E) Two consecutive transport cycles. TmrAB reconstituted in liposomes was incubated with C4$^{ATTO655}$ peptide, ATP/ADP, MgCl$_2$, and CP$^F$s as described in (A) for 5 min at 45℃, 60 min at 4℃, 5 min at 45℃, and 60 min at 4℃. Transported peptides per liposome were evaluated as described in (B) and corrected by background levels in the presence of ADP. In (B–E), mean values ± SD (n = 3) are displayed.

The online version of this article includes the following source data and figure supplement(s) for figure 6:

**Source data 1.** Source data for *Figure 6*.

**Figure supplement 1.** Linear regression of quantitative flow cytometry analysis.

**Figure supplement 1—source data 1.** Source Data for *Figure 6—figure supplement 1*.

TmrAB were incubated with C4[ATTO655] peptide, ATP/ADP, and CP[F]s for 5 min at 45°C to induce the IF-to-OF switch and for 60 min at 4°C to allow the OF-to-IF return transition. Both steps were repeated to yield two consecutive cycles of conformational transitions. In the presence of CP13[F], peptide transport was exclusively observed along the first IF-to-OF switch, indicating a conformational arrest of wild-type TmrAB over a period of 2 hr (*Figure 6*). For CP14[F], peptide transport was detected in both IF-to-OF switches, which confirmed the slow relaxation of wild-type TmrAB from a pre-hydrolysis to an IF state between both cycles (*Figure 6E*). In the presence of CP6[F] or CP12[F], peptide translocation was completely abrogated as the underlying IF-to-OF switch was precluded (*Figure 6E*, *Figure 6—figure supplement 1*). Taken together, the macrocyclic peptides CP13 and CP14 stabilize a pre-hydrolysis state, thus allowing a single-turnover translocation along the IF-to-OF switch.

## Discussion

In this study, we identified macrocyclic peptides as multimode inhibitors, which block the heterodimeric ABC transport complex TmrAB at different stages in the substrate translocation cycle with nanomolar affinities. We reveal that CPs can be employed to arrest TmrAB at different stages during a single IF-to-OF switch induced by ATP binding, which drives substrate transport across the membrane (*Figure 7*). These IF-to-OF and OF-to-IF inhibitors stabilize TmrAB states that are distinct from the vanadate trapped states and allowed us to unravel mechanistic principles of substrate transport for wild-type ABC transproters and illustrate the potential of selective inhibitors as next-generation antibiotics. These findings will help to investigate key functional determinants of membrane proteins involved in multidrug resistance.

To select specific CPs for a heterodimeric ABC transport complex, we successfully applied the RaPID system to isolate high-affinity CPs for soluble and membrane proteins (*Yu et al., 2017*; *Sakai et al., 2019*; *Kodan et al., 2014*; *Tanaka et al., 2013*). It has been shown that CPs are powerful agents to stabilize protein conformers or protein complexes for structural and functional investigations (*Huang et al., 2019*; *Passioura and Suga, 2017*). Upon cyclization of the peptide backbone, CPs mimic the complementary-determining regions (CDRs) of antibodies penetrating into buried cavities (*Hill et al., 2014*; *McGeary and Fairlie, 1998*). Compared to conventional antibodies and antibody fragments, the incorporation of unnatural amino acids largely expands the chemical

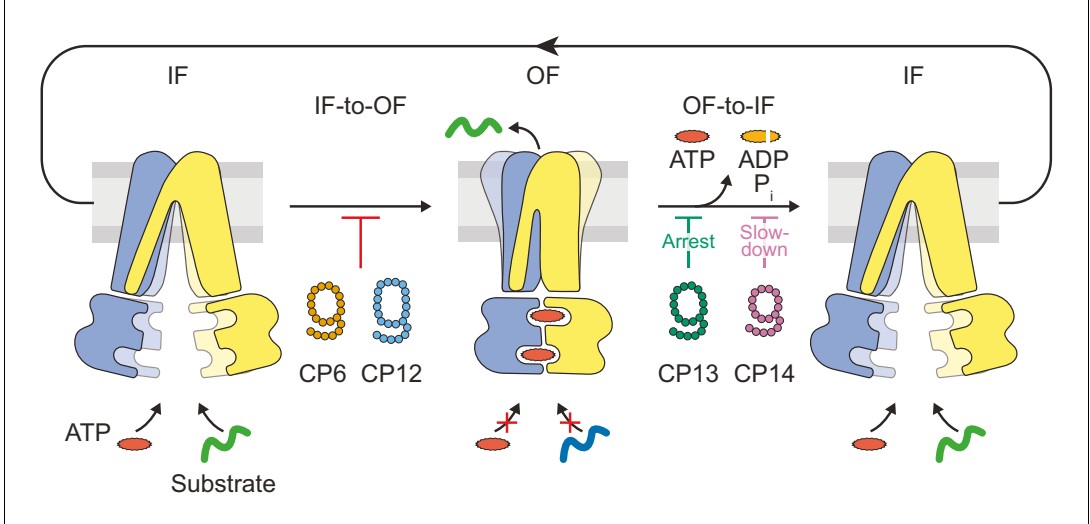

**Figure 7.** Substrate translocation precedes ATP hydrolysis in a heterodimeric ABC transporter. In the resting IF state, TmrAB binds nucleotides and substrates independently, which is not affected by CPs. ATP binding induces an IF-to-OF switch, which drives unidirectional substrate translocation. CP6[F] and CP12[F] bind preferentially to the IF state and block the transition to the OF state, preventing ATP occlusion and ATP hydrolysis. CP13[F] and CP14[F] favor and stabilize a pre-hydrolysis OF state after the IF-to-OF conformation switch and peptide translocation. CPs block ATP hydrolysis at different steps of the transport cycle. In the absence of CPs, ATP hydrolysis and phosphate release initiate the OF-to-IF return restoring transporter function.

diversity of paratopes to reach unexplored binding modes (*Craik et al., 2013*; *Makley and Gestwicki, 2013*; *Tsomaia, 2015*). By using the catalytically reduced variant TmrA$^{EQ}$B reconstituted in lipid nanodiscs as bait during selections, we hypothesized that even transient states could be populated that would allow us to isolate macrocycles with unique bioactivities (*Hofmann et al., 2019*; *Stefan et al., 2020*). As demonstrated by affinity isolation, gel filtration, and native mass spectrometry, all CPs selected against TmrA$^{EQ}$B in lipid nanodiscs also specifically bind to wild-type TmrAB in various lipid models. These findings illustrate the potential of the RaPID system to identify specific binders for delicate membrane proteins and less-populated transition states under carefully chosen conditions, which is a key requirement for pharmaceutical compounds (*Vinogradov et al., 2019*; *Zorzi et al., 2017*). Intriguingly, the most enriched macrocyclic peptides also display inhibitory activity, even though the selection is based on target binding alone. Based on deep sequencing, CP6 and CP12, acting on the IF state, showed higher enrichment than CP13 and CP14, acting on the OF state, which is consistent with the selections being performed at 4°C, where TmrAB adopts the IF state. However, due to high binding affinities of OF-specific CP13 and CP14, these CPs were also enriched along the course of the selection. In contrast to CPs elicited against the homodimeric ABC transporter cmABCB1 from *Cyanidioschyzon merolae* (aCAP) (*Kodan et al., 2014*) and the MATE transporter (MaD3S, MaD5, MaL6) (*Tanaka et al., 2013*), the CPs discovered in this study exclusively target the transporter from the cytosolic side. In the case of aCAP, which was selected against detergent stabilized P-glycoprotein, the CP not only interacts with both protein subunits but also with the membrane, acting as an allosteric inhibitor thus stabilizing this ABC transporter in the IF conformation. Like aCAP, the CPs from this study are also enriched in aliphatic and aromatic amino acids, which could indicate membrane interactions. The ability of the CP6, CP12, CP13, and CP14 to specifically interact with either the IF or OF TmrAB conformer however, again suggests that the success of this selection experiment in discovering multimode inhibitors relied on specific preparation of the bait protein during the selection process.

In recent years, the molecular understanding of the conformation space of ABC transporters has been largely expanded (*Thomas and Tampé, 2020*; *Locher, 2016*; *Hofmann et al., 2019*; *Fan et al., 2020*). High-resolution structures in different conformational states allowed to build a detailed mechanistic model of the substrate transport cycle composed of two major conformational transitions (IF-to-OF and OF-to-IF) (*Hofmann et al., 2019*; *Fan et al., 2020*; *Manolaridis et al., 2018*; *Kim and Chen, 2018*). We demonstrated unidirectional substrate transport along a single IF-to-OF switch induced by ATP binding for a catalytically reduced variant of TmrAB (*Stefan et al., 2020*). These results were consistent with the processive-clamp/ATP switch model where ATP binding and the subsequent IF-to-OF switch represent the power stroke to drive substrate transport (*Higgins and Linton, 2004*; *Janas et al., 2003*; *Abele and Tampé, 2004*). To date, however, it remains an open question whether a single ATP-binding-induced conformational switch drives substrate translocation in the case of fully functional wild-type ABC transporters. Additionally, the relevance of futile ATP hydrolysis cycles as well as factors contributing to energy coupling (ratio of consumed ATP per translocated substrate) have remained unexplained for wild-type TmrAB and other ABC systems (*Fan et al., 2020*; *Bock et al., 2019*). To address these pressing questions, we used the in vitro-selected peptidic macrocycles as mechanistic probes to arrest TmrAB at different stages of the substrate transport cycle. Inhibitors for ABC transporters exploit various modes of action such as substrate competition, conformational arrest, or hampering of ATP hydrolysis to affect substrate transport (*Srikant and Gaudet, 2019*; *Aller et al., 2009*; *Ozvegy-Laczka et al., 2005*; *Orlando and Liao, 2020*; *Ho et al., 2018*).

For the selected CPs, we observed a remarkably strong inhibition of ATP hydrolysis and substrate transport, which did not largely interfere with ATP or substrate binding. We rationalized that these macrocyclic peptides affect conformational transitions within the translocation cycle. Using nucleotide occlusion as readout, we could show that CP13$^F$ and CP14$^F$ stabilize wild-type TmrAB in a prehydrolysis state, unlike the vanadate-trapped state solved by cryo-EM, which occludes ADP and vanadate in the canonical ATP-binding site (*Hofmann et al., 2019*). Our hypothesis was that wild-type TmrAB in the presence of CP13 or CP14 is able to perform the the IF-to-OF transition induced by ATP binding, whereas only the opposite OF-to-IF transitions triggered by ATP hydrolysis are blocked. To test this hypothesis, we performed peptide transport assays with fully functional wild-type TmrAB in the presence of CP13 and CP14 and determined stoichiometric or near-stoichiometric amount of transported peptide per TmrAB complex. This coupling rate is comparable to the almost

ideal coupling ratio observed under single-turnover conditions for a catalytically reduced variant (*Stefan et al., 2020*). This energy coupling is substantially higher compared to multiple-turnover steady-state approaches for wild-type TmrAB (*Bock et al., 2019*).

By analyzing peptide translocation of arrested wild-type TmrAB over long periods of time, we observed a slow turnover in the case of CP14, and a full conformational arrest for CP13. These observations were underpinned by analyzing two consecutive cycles of IF-to-OF switches and OF-to-IF return transitions in regard to peptide translocation. In the presence of CP14, wild-type TmrAB was able to transport peptides along the ATP-induced IF-to-OF switch followed by the relaxation from an ATP-bound OF state to an IF state in order to start a new IF-to-OF switch in the second cycle. In contrast, after the translocation of peptide, CP13 arrested wild-type TmrAB in an ATP-bound state and prevented the OF-to-IF return transition and further conformational cycling. On the other hand, CP6 and CP12 block peptide transport to background level as an IF-to-OF transition and ATP occlusion is prevented. Using macrocyclic inhibitors, we demonstrated, for the first time for a wild-type ABC transporter, that substrate translocation is driven along a single IF-to-OF switch triggered by ATP binding.

In this work, we utilized synthetic macrocyclic inhibitors to reveal fundamental mechanistic principles of ATP binding and hydrolysis, conformational transitions, and energy coupling of substrate translocation of a heterodimeric ABC transport complex. We highlight the versatility of macrocycles and their potential to study mechanistic determinants of membrane protein complexes. In future approaches, conformation-selective compounds along with single-liposome or single-molecule techniques will be essential to uncover unexplored aspects of conformational transitions, translocation events, and energy coupling of membrane translocation complexes.

# Materials and methods

## Key resources table

| Reagent type (species) or resource | Designation | Source or reference | Identifiers | Additional information |
|---|---|---|---|---|
| Antibody | Monoclonal α-His antibody | Sigma-Aldrich | SAB1305538 | Mouse origin. Final dilution: 1/2,000 (v/v) |
| Antibody | α-Mouse-HRP conjugate | Sigma-Aldrich | AP130P | Goat origin. Final dilution: 1/20,000 (v/v) |
| Chemical compound, drug | β-n-Dodecyl β-D-maltoside (DDM) | Carl Roth | CN26.5 | |
| Chemical compound, drug | Bovine brain lipid extract | Sigma-Aldrich | B1502 | |
| Chemical compound, drug | [2,5′,8-$^3$H(N)]-ATP ($^3$H-ATP) | PerkinElmer | NET118900 | |
| Chemical compound, drug | [α$^{32}$P]-ATP | Hartmann Analytic | FP-207 | |
| Chemical compound, drug | Copper-chelated PVT SPA beads | PerkinElmer | RPNQ0095 | |
| Chemical compound, drug | ClAc-D-Tyr-CME | Synthesized according to DOI: 10.1021/cb200388k | | |
| Chemical compound, drug | Fmoc-protected amino acids | Merck Millipore/Watanabe Chemical Industries | various | |
| Chemical compound, drug | HBTU | Watanabe Chemical Industries | A00149 | |
| Chemical compound, drug | HOBt | Watanabe Chemical Industries | A00014 | |
| Chemical compound, drug | NovaPEG Rink Amide resin | Merck Millipore | 855047 | |

*Continued on next page*

*Continued*

| Reagent type (species) or resource | Designation | Source or reference | Identifiers | Additional information |
|---|---|---|---|---|
| Chemical compound, drug | N,N-Diisopropylethylamine | Nacalai Tesque | 14014–55 | |
| Chemical compound, drug | N,N-Dimethylformamide | Nacalai Tesque | 13016–23 | |
| Chemical compound, drug | 5/6-Carboxyfluorescein succinimidyl ester | Thermo Fisher Scientific | 46410 | |
| Chemical compound, drug | Acetonitrile | Wako Chemicals | 015–08633 | |
| Chemical compound, drug | Trifluoroacetic acid | Nacalai Tesque | 3483305 | |
| Chemical compound, drug | D-Biotin | Nacalai Tesque | 04822–91 | |
| Chemical compound, drug | Pluronic F127 | Sigma-Aldrich | P2443 | |
| Chemical compound, drug | Albumin, Bovine, Acetylated | Nacalai Tesque | 01278–44 | |
| Chemical compound, drug | NTPs | Jena Bioscience | NU-1010 NU-1011 NU-1012 NU-1013 | |
| Chemical compound, drug | Dynabeads M-280 Streptavidin | Thermo Fisher Scientific | 11206 | |
| Sequence-based reagent | T7g10M.F46 | Eurofins Genomics K.K. (Japan) | PCR primer | TAATACGACTCACTATAGGGTTAACTTTAAGAAGGAGATATACATA |
| Sequence-based reagent | NNK(n).R(3n + 45) n = 10–15 | Eurofins Genomics K.K. (Japan) | PCR primer for DNA library | GCTGCCGCTGCCGCTGCCGCA(MNN)$_n$ CATATGTATATCTCCTTCTTAAAG |
| Sequence-based reagent | CGS3an13.R36 | Eurofins Genomics K.K. (Japan) | PCR primer | TTTCCGCCCCCCGTCCTAGCTGCCGCTGCCGCTGCC |
| Sequence-based reagent | Ini-3'.R20-Me | Gene Design Inc (Japan) | PCR primer for tRNA$^{fMet}_{CAU}$ assembly | TGmGTTGCGGGGGCCGGATTT (Gm = 2'-Methoxylated G) |
| Sequence-based reagent | Ini-3'.R38 | Eurofins Genomics K.K. (Japan) | PCR primer for tRNA$^{fMet}_{CAU}$ assembly | TGGTTGCGGGGGCCGGATTTGAACCGACGATCTTCGGG |
| Sequence-based reagent | Ini1-1G-5'.F49 | Eurofins Genomics K.K. (Japan) | PCR primer for tRNA$^{fMet}_{CAU}$ assembly | GTAATACGACTCACTATAGGCGGGGTGGAGCAGCCTGGTAGCTCGTCGG |
| Sequence-based reagent | Ini cat.R44 | Eurofins Genomics K.K. (Japan) | PCR primer for tRNA$^{fMet}_{CAU}$ assembly | GAACCGACGATCTTCGGGTTATGAGCCCGACGAGCTACCAGGCT |
| Sequence-based reagent | Fx5'.F36 | Eurofins Genomics K.K. (Japan) | PCR primer for eFx assembly | GTAATACGACTCACTATAGGATCGAAAGATTTCCGC |
| Sequence-based reagent | eFx.R45 | Eurofins Genomics K.K. (Japan) | PCR primer for eFx assembly | ACCTAACGCTAATCCCCTTTCGGGGCCGCGGAAATCTTTCGATCC |
| Sequence-based reagent | eFx.R18 | Eurofins Genomics K.K. (Japan) | PCR primer for eFx assembly | ACCTAACGCTAATCCCCT |
| Sequence-based reagent | T7e × 5 .F22 | Eurofins Genomics K.K. (Japan) | PCR primer for eFx assembly | GGCGTAATACGACTCACTATAG |
| Sequence-based reagent | DNA-PEG-puromycin | Gene Design Inc, Osaka, Japan | linker for mRNA display | CTCCCGCCCCCCGTCC-(PEG18)5-CC-Pu |
| Gene | TmrA | Q72J05 | TTC0976 | Species: *Thermus thermophilus* |
| Gene | TmrB | Q72J04 | TTC0977 | Species: *Thermus thermophilus* |
| Peptide, recombinant protein | RRY-C*-KSTEL | This study (methods and material) | | C* denotes fluorescein-labeled Cys |

*Continued on next page*

*Continued*

| Reagent type (species) or resource | Designation | Source or reference | Identifiers | Additional information |
|---|---|---|---|---|
| Peptide, recombinant protein | Macrocyclic peptides CP6, CP12, C13 and CP14 | This study (methods and material) | | |
| Peptide, recombinant protein | KOD DNA Polymerase | Prepared in house (methods and material) | | |
| Peptide, recombinant protein | T7 RNA polymerase | Prepared in house (methods and material) | | |
| Peptide, recombinant protein | T4 RNA ligase | Prepared in house (methods and material) | | |
| Peptide, recombinant protein | FIT system | Prepared in house according to DOI: 10.1038/nprot.2015.082 | | 50 mM HEPES-KOH (pH 7.6), 12 mM magnesium acetate, 100 mM potassium acetate, 2 mM spermidine, 20 mM creatine phosphate, 2 mM DTT, 2 mM ATP, 2 mM GTP, 1 mM CTP, 1 mM UTP, 0.5 mM 19 proteinogenic amino acids other than Met, 1.5 mg/ml *E. coli* total tRNA, 0.73 µM AlaRS, 0.03 µM ArgRS, 0.38 µM AsnRS, 0.13 µM AspRS, 0.02 µM CysRS, 0.06 µM GlnRS, 0.23 µM GluRS, 0.09 µM GlyRS, 0.02 µM HisRS, 0.4 µM IleRS, 0.04 µM LeuRS, 0.11 µM LysRS, 0.03 µM MetRS, 0.68 µM PheRS, 0.16 µM ProRS, 0.04 µM SerRS, 0.09 µM ThrRS, 0.03 µM TrpRS, 0.02 µM TyrRS, 0.02 µM ValRS, 0.6 µM MTF, 2.7 µM IF1, 0.4 µM IF2, 1.5 µM IF3, 0.26 µM EF-G, 10 µM EF-Tu, 10 µM EF-Ts, 0.25 µM RF2, 0.17 µM RF3, 0.5 µM RRF, 0.1 µM T7 RNA polymerase, 4 µg/ml creatine kinase, 3 µg/ml myokinase, 0.1 µM pyrophosphatase, 0.1 µM nucleotide-diphosphatase kinase, 1.2 µM ribosome |
| Recombinant DNA reagent | pET-22b | Merck Millipore | 69744 | Vector for protein expression in *E. coli* |
| Strain, strain background (*Escherichia coli*) | BL21(DE3) | Thermo Fisher | C600003 | Chemically competent cells |
| Software, Algorithm | Prism 5 | GraphPad | | |
| Software, algorithm | Cytoscape | Shannon P et al. Genome Research 2003 13(11) 2498–504 | | |
| Software, algorithm | EFI-EST | Gerlt JA et al. Biochim Biophys Acta 1854: 1019-37 | | |
| Software, algorithm | WebLogo | Crooks GE et al. Genome 561 Res 14: 1188–90 | | |

## Expression and purification of TmrAB

TmrAB and TmrA$^{E523Q}$B (TmrA$^{EQ}$B), harboring a C-terminal His-tag at TmrA, were expressed in *E. coli* BL21(DE3) grown in LB high-salt media at 37 ℃ (*Nöll et al., 2017*; *Hofmann et al., 2019*; *Stefan et al., 2020*). Expression was induced at an OD$_{600}$ of 0.6 by adding 0.5 mM isopropyl β-D-thiogalactopyranoside (IPTG, Carbolution) for 3 hr at 37℃. Cells were harvested by centrifugation for 15 min at 4,500 g and resuspended in lysis buffer (20 mM HEPES-NaOH pH 7.5, 300 mM NaCl, 50 µg/ml lysozyme, 0.2 mM PMSF). After sonication, membranes were isolated by centrifugation at 100,000 g for 30 min at 4℃. Membranes were solubilized by adding 20 mM β-n-dodecyl β-D-maltoside (β-DDM, Carl Roth) in purification buffer (20 mM HEPES-NaOH pH 7.5, 300 mM NaCl) for 1 hr at 4℃. Samples were cleared at 100,000 g for 30 min, and solubilized proteins were loaded onto Ni-

NTA agarose (Qiagen) at 4°C for 1 hr. Resin was washed with 20 column volumes of SEC buffer (20 mM HEPES-NaOH pH 7.5, 150 mM NaCl, 1 mM β-DDM) containing 50 mM imidazole. TmrAB was eluted in SEC buffer containing 300 mM imidazole. The eluate buffer was exchanged to SEC buffer using PD-10 desalting column (GE Healthcare).

## Expression and purification of membrane scaffold proteins

MSP1D1 was expressed in *E. coli* BL21(DE3) cultivated in LB high-salt media at 37°C. At an $OD_{600}$ of 1.0, expression was induced by adding 1 mM IPTG for 1 hr at 37°C. Temperature was reduced to 28°C for additional 4 hr. MSP1D1 was purified as described (*Hofmann et al., 2019*). Briefly, cells were harvested by centrifugation at 4,500 g for 15 min and disrupted by sonication in 40 mM Tris–HCl pH 8.0, 300 mM NaCl, 1% Triton X-100. Lysate was cleared by centrifugation for 30 min at 30,000 g and loaded onto Ni-NTA agarose (Qiagen) for 1 hr at 4°C. Resin was washed with 20 column volumes of 40 mM Tris–HCl pH 8.0, 300 mM NaCl, 50 mM imidazole. MSP1D1 was eluted in 40 mM Tris–HCl pH 8.0, 300 mM NaCl, 400 mM imidazole. The eluate buffer was exchanged to 20 mM Tris–HCl pH 7.4, 100 mM NaCl, 0.5 mM EDTA using PD-10 desalting column (GE Healthcare).

## Reconstitution in lipid nanodiscs

TmrAB was reconstituted in lipid nanodiscs (*Hofmann et al., 2019*) composed of biotinylated MSP1D1. For biotinylation, purified MSP1D1 was buffer-exchanged to 20 mM HEPES-NaOH pH 7.5, 150 mM NaCl using Zeba spin desalting column (Thermo Fisher). A four-fold molar excess of EZ-Link $NHS-PEG_4$-biotin (Thermo Fisher) was added for 2 hr on ice. Residual $NHS-PEG_4$-biotin was quenched by adding 10 mM of Tris–HCl pH 8.0. To remove the excess of $NHS-PEG_4$-biotin, samples were buffer exchanged to 20 mM HEPES-NaOH pH 7.5, 150 mM NaCl using PD-10 desalting column (GE Healthcare). Bovine brain lipids (Sigma-Aldrich) were solubilized in 20 mM of β-DDM. Biotinylated MSP1D1 was mixed with bovine brain lipids with and without TmrAB in a TmrAB/MSP1D1/lipid molar ratio of 1/7.5/100 in SEC buffer without detergent to form TmrAB-filled or empty liposomes, respectively. Samples were incubated for 30 min at 20°C, and SM-2 Bio-beads (Bio-Rad) were added in two steps at 4°C (1 hr and overnight) for detergent removal. Samples were concentrated using Amicon Ultra-0.5 ml centrifugal filters with 50 kDa cut-off (Merck Millipore). Reconstituted nanodiscs were isolated by size-exclusion chromatography (SEC) via Superdex 200 Increase 3.2/300 (GE Healthcare).

## Library generation

For library construction, DNA oligonucleotides (Eurofins Genomics) were assembled by PCR amplification using KOD polymerase in Phusion buffer (NEB). DNA products were purified by phenol/chloroform/isoamylalcohol extraction and ethanol precipitation (*Hayashi et al., 2012*). The final sequence contained a T7 promotor, the start codon, the randomized region flanked by a cysteine codon, a $(GS)_3$ linker, the stop codon, and a short sequence stretch for annealing of the polyethylene glycol (PEG)-puromycin splinter. Templates for $tRNA_{fMet}^{CAU}$ and Flexizyme (eFx) were prepared by primer assembly and converted to RNA through run-off transcription using T7 RNA polymerase (*Goto et al., 2008*; *Shin et al., 2017*). The $tRNA_{fMet}^{CAU}$ was aminoacylated with chloroacetyl-D-tyrosine-cyanomethyl ester (ClAc-D-Tyr-CME) by eFx for 2 hr on ice (*Goto et al., 2011*; *Hayashi et al., 2012*).

## Selection of macrocyclic peptides

Macrocyclic peptides against the heterodimeric ABC transporter were selected using the RaPID system as described (*Yamagishi et al., 2011*; *Hayashi et al., 2012*). Briefly, DNA libraries ($NNK_{10-15}$) were transcribed to RNA using T7 RNA polymerase and purified by denaturing PAGE and ethanol precipitation. Libraries were mixed in equimolar ratios, and a DNA-PEG-puromycin splinter was covalently attached using T4 RNA ligase. The product was purified by phenol/chloroform/isoamylalcohol extraction and ethanol precipitation (*Hayashi et al., 2012*). Puromycin-modified RNAs were translated by FIT supplemented with 50 µM $ClAc-D-Tyr-tRNA_{fMet}^{CAU}$ for 30 min at 37°C at an RNA-puromycin concentration of 1 µM. The used FIT system was deficient in RF1, L-methionine, and 10-formyl-5,6,7,8-tetrahydrofolic acid to allow reprogramming of the initiator position. Subsequently, EDTA (10 mM) was added, followed by reverse transcription using MMLV reverse transcriptase (Promega).

Dynabeads M280 Streptavidin (Invitrogen) were loaded with $TmrA^{EQ}B$ reconstituted in biotinylated lipid nanodiscs (binding capacity: 0.3 pmol/μl bead slurry) for 20 min at 4°C. Beads were washed with selection buffer (20 mM HEPES-NaOH pH 7.5, 150 mM NaCl, 0.001% Pluronic F127) supplemented with acetylated BSA (1 g/l, Nacalai Tesque). Subsequently, the mRNA-peptide library was added and incubated for 15 min at 4°C (without supplementing additional reducing agents). The beads were washed three times with selection buffer followed by elution of cDNA in Phusion buffer for 5 min at 95°C. Recovered cDNA was quantified by real-time PCR using Taq polymerase and SYBR Green and amplified by Fusion polymerase (NEB). In subsequent rounds, the mRNA-peptide library was subjected to unmodified beads and beads with immobilized empty nanodiscs, to remove non-target specific binders prior to incubation with $TmrA^{EQ}B$ (*Hayashi et al., 2012*). Isolated and amplified DNA served as the input for subsequent selection rounds in an iterative manner. For sequencing, Nextera XT dual indices (Illumina) were added to both DNA termini of the recovered DNA pool by PCR using KOD polymerase. Sequencing was performed on a MiSeq sequencer (Illumina) (*Rogers et al., 2018*).

## Synthesis of macrocyclic peptides

Macrocycles were produced by Fmoc solid-phase synthesis on a Syro I (Biotage) synthesizer. Nova-PEG Rink amide resin (Merck Millipore) was used as the solid support, giving rise to C-terminal amides. A combination of HBTU/HOBt (1/1, six equivalents) was utilized for carboxylate activation during the coupling steps. Double couplings were performed for Arg residues. To allow on-resin labeling and cyclization, orthogonally protected amino acids – Fmoc-Lys(MMT)-OH and Fmoc-Cys(StBu)-OH – were used, respectively. Chloroacetylation of the N terminus was achieved by manual coupling of chloroacetic acid using HBTU/HOBt (1/1, 10 equivalents) on resin for 1 hr. For site-directed labeling, the MMT protection group of the C-terminal lysine was cleaved by repeated incubations with 1% (v/v) trifluoroacetic acid (TFA) and 5% (v/v) triisopropyl silane (TIPS) in dichloromethane (DCM). The resin was washed with DCM, N,N-dimethylformamide (DMF), 20% (v/v) N,N-diisopropylethylamine in DMF, DMF, methanol, and DCM. Deprotected lysine side chains were conjugated with 5/6-carboxyfluorescein succinimidyl ester (ThermoFisher Scientific) or D-biotin in combination with HBTU/HOBt. The resin was dried and incubated with TFA/2,2′-(ethylenedioxy) diethanethiol (DODT)/TIPS/water in a 92.5/2.5/2.5/2.5 (v/v) ratio for 2.5 hr at 20°C. Excess TFA was removed using GeneVac centrifugal evaporators (SP Industries), and peptides were precipitated and washed six times with diethylether on ice. Peptides were dissolved in 90% (v/v) dimethylsulfoxide (DMSO) in water, and the pH was adjusted to 10.0 by addition of triethylamine. For complete macro-cyclization, samples were incubated at 20°C for 2 hr. TFA was added to reduce the pH to a value between 1.0 and 2.0. For $CP6^F$, the StBu protection group of Cys was removed by adding 20 equivalents of tributylphosphine in 95% (v/v) trifluoroethanol in water. $CP6^F$ was pre-purified by reversed-phase (RP) HPLC using a SNAP $C_{18}$ column (Biotage) on an Isolera-One system (Biotage), applying a linear acetonitrile gradient containing 0.1% (v/v) TFA. All macrocycles were purified by RP-HPLC on a Prominence LC-20AP system (Shimadzu) equipped with a Chromolith column (Merck Millipore) applying a linear acetonitrile gradient containing 0.1% (v/v) TFA. Identity and purity of macrocycles was confirmed by MALDI-TOF-MS and HPLC analysis.

## Reconstitution in liposomes

Liposomes composed of *E. coli* polar lipids and DOPC (7/3 w/w; Anatrace) were destabilized with Triton X-100, and purified TmrAB was added in a 1/20 (w/w) ratio. Detergent was removed by adding SM-2 Bio-beads (Bio-Rad) in four steps (1 hr, overnight, 2 hr, and 2 hr) at 4°C. Proteoliposomes were harvested at 270,000 g for 30 min at 4°C and resuspended in transport buffer (20 mM HEPES-NaOH pH 7.5, 150 mM NaCl, 5% [v/v] glycerol) to a final lipid concentration of 5 mg/ml.

## Fluorescence anisotropy

To determine equilibrium binding constants, $CP^F$s (50 nM) were incubated with increasing concentrations of TmrAB-containing or empty liposomes. To probe the conformational selectivity, $CP^F$s (50 nM) were mixed with detergent-solubilized IF TmrAB or OF $TmrA^{EQ}B$. Before, OF $TmrA^{EQ}B$ was nucleotide-trapped by incubation with ATP (1 mM) and $MgCl_2$ (3 mM) in the absence of $CP^F$s for 5 min at 45°C. For competition binding studies, $CP^F$s (50 nM) were mixed with detergent-solubilized

TmrAB (0.6 µM for CP6$^F$ and CP13$^F$, 0.7 µM for CP12$^F$, 0.5 µM for CP14$^F$) in the presence of RRY-C$^{ATTO655}$KSTEL (C4$^{ATTO655}$, 2 µM) or RRYQKSTEL peptide (R9LQK, 200 µM). For peptide competition studies, RRYC$^{fluorescein}$KSTEL (C4F, 50 nM) were incubated with detergent-solubilized TmrAB (4 µM), CP$^B$s (6 µM), or C4$^{ATTO655}$ (10 µM). To examine conformational selective binding, TmrAB (0.6 µM for CP6$^F$ and CP13$^F$, 0.7 µM for CP12$^F$, 0.5 µM for CP14$^F$) were incubated with CP$^F$s (50 nM) and SDS (1% (v/v)) for 10 min at 20°C. Fluorescence anisotropy was analyzed at $\lambda_{ex/em}$ of 485/520 nm using a microplate reader (CLARIOstar, BMG LABTECH). Fluorescence anisotropy was calculated using:

$$r = \frac{I_{||} - I_{\perp}}{I_{||} + 2 \cdot I_{\perp}} \tag{1}$$

with $I_{||}$: parallel intensity and $I_{\perp}$: perpendicular intensity. Binding of CP$^F$s were fitted by a one-site binding model (Langmuir 1:1) using:

$$Y = \frac{B_{max} \cdot [L]}{K_D + [L]} \tag{2}$$

with Y: binding, $B_{max}$: maximal binding, $K_D$: equilibrium dissociation constant, and [L]: ligand concentration. Data represent mean ± SD from three experiments.

## Macrocyclic peptide-mediated isolation of TmrAB

CP$^B$s (1 µM) were immobilized on streptavidin beads (Thermo Scientific) and incubated with purified TmrAB for 1 hr at 4°C. Alternatively, DDM-solubilized *E. coli* membranes containing TmrAB were added to reach a final protein concentration of 2 mg/ml. Beads were washed by centrifugation at 200 g for 1 min with SEC buffer (20 mM HEPES-NaOH pH 7.5, 150 mM NaCl, 1 mM β-DDM). Bound protein was eluted by adding SDS loading buffer (50 mM Tris–HCl pH 6.8, 2% [w/v] SDS, 10% [v/v] glycerol, 0.1% [w/v] bromophenol blue, 100 mM DTT) for 10 min at 95°C. Equal aliquots of input, flow-through, washing fraction, and eluate were analyzed by SDS–PAGE and immunoblotting (α-His).

## Gel filtration

Purified TmrAB (1 µM) was incubated with CP12$^F$, CP13$^F$, CP14$^F$ (2 µM each), or CP6$^F$ (5 µM) in SEC buffer (20 mM HEPES-NaOH pH 7.5, 150 mM NaCl, 1 mM β-DDM) for 10 min on ice. CP$^F$-TmrAB complexes were analyzed by gel filtration using Superdex 200 Increase 3.2/300 (GE Healthcare).

## Mass spectrometry

Detergent-solubilized TmrAB was buffer-exchanged to ESI buffer (100 mM ammonium acetate pH 7.2, 1 mM β-DDM) using Zeba desalting columns (Thermo Scientific). TmrAB (4 µM) was incubated with a twofold molar excess of CP$^F$s for 10 min on ice. CP$^F$-TmrAB complexes were examined by electrospray ionization-mass spectrometry (ESI-MS) on a Synapt G2S (Waters Corporation) equipped with a high-mass quadrupole upgrade. Pd/Pt sputtered nESI tips were pulled in house from borosilicate glass capillaries on a Flaming/Brown micropipette puller (P-1000, Sutter Instrument Co). Capillary and cone voltages were set to 1.85 kV and 150 V, respectively. The source block temperature was set to 30°C. Neither trap nor transfer collision energy was used. Data analysis was performed using the software MassLynx V4.1.

## ATP binding assay

DDM-solubilized TmrAB (0.2 µM) was incubated with ATP (2.8 µM), [2,5′,8-$^3$H(N)]-ATP (0.2 µM, $^3$H-ATP, PerkinElmer), MgCl$_2$ (3 mM), and CP$^B$s (1 µM) for 30 min on ice. Copper-chelated SPA beads (PerkinElmer) were added to a final concentration of 5 mg/ml. Total binding was determined at 20°C in cpm mode (Wallac MicroBeta). Bound TmrAB complexes were eluted by adding 200 mM of imidazole, and background binding was determined. Data represent mean ± SD from three experiments.

## Peptide transport assays

TmrAB in liposomes (0.4 µM) were incubated with ATP/ADP (3 mM), MgCl$_2$ (5 mM), C4F peptide (3 µM), and CP$^F$s (1 µM) for 15 min at 45°C. Reactions were stopped by adding EDTA (10 mM), and

proteoliposomes were washed with transport buffer (20 mM HEPES-NaOH pH 7.5, 150 mM NaCl, 5% [v/v] glycerol) on PEI-equilibrated MultiScreen plates (0.65 μm, Merck Millipore). Proteoliposomes were solubilized by adding SDS (1% [v/v]) in transport buffer for 10 min at 20°C. Transported peptides were quantified at $\lambda_{ex/em}$ = 485/520 nm using a microplate reader (CLARIOstar, BMG LABTECH).

## Single-liposome assays

TmrAB-containing liposomes (0.4 μM) were incubated with $C4^{ATTO655}$ peptides (1 μM), ATP or ADP (3 mM each), $MgCl_2$ (5 mM), and $CP^F$s (1 μM) for 5 min at 45°C. Transport reactions were stopped by adding EDTA (10 mM). Samples were diluted to a final TmrAB concentration of 20 nM with transport buffer (20 mM HEPES-NaOH pH 7.5, 150 mM NaCl, 5% [v/v] glycerol) and analyzed by flow cytometry (FACSCelesta). For regression analysis, liposomes were loaded with increasing amounts of peptides to convert the fluorescence intensities into the number of peptides per liposome (*Stefan et al., 2020*). For this, liposomes were destabilized by adding Triton X-100 while increasing the $C4^{ATTO655}$ concentration. Equal amounts of carboxyfluorescein (Sigma-Aldrich) served as loading control. Detergent was removed by adding SM-2 Bio-beads (Bio-Rad) as detailed above. Liposomes were harvested for 30 min at 270,000 g and washed three times by centrifugation. Generally, 20,000–100,000 proteoliposomes were selected according to sideward and forward scatter areas. Single events were selected based on the height of forward scatter plotted against the area of forward scatter. Mean fluorescence intensities of fluorescein and ATTO655 were calculated using FlowJo 10.6.1 software. Data represent mean ± SD from three experiments.

## ATP hydrolysis assays

TmrAB reconstituted in liposomes (100 nM) were incubated with ATP (2 mM) traced with [$\gamma^{32}$P]-ATP (Hartmann Analytic), ouabain (1 mM), $NaN_3$ (5 mM), EGTA (50 μM), $MgCl_2$ (3 mM), and $CP^F$s (1 μM) for 10 min at 45°C. Autohydrolysis and background hydrolysis were examined in the absence of proteoliposomes or the presence of EDTA (10 mM). Samples were spotted onto polyethyleneimine cellulose plates (Merck Millipore), and thin-layer chromatography was performed with 0.8 M LiCl-acetic acid pH 3.2. Plates were developed overnight on Exposure Cassette-K (Bio-Rad) and evaluated on Personal Molecular Imager System (Bio-Rad). Data were recorded in triplicates and the mean values ± SD (n = 3) are shown.

## Nucleotide occlusion

Detergent-solubilized TmrAB (2 μM) were incubated with ATP (1 mM) traced with [$\alpha^{32}$P]-ATP (Hartmann Analytic), $MgCl_2$ (5 mM), $CP^F$s (4 μM), or orthovanadate (1 mM) for 5 min at 45°C. Cold ATP (10 mM) was added, and unbound nucleotides were removed by rapid gel filtration (Bio-Spin columns P-30, Bio-Rad). ATP (10 mM) was added, and samples were spotted onto polyethyleneimine cellulose plates (Merck Millipore). Thin-layer chromatography was performed using 0.75 M $KH_2PO_4$ pH 3.4. Plates were developed overnight on Exposure Cassette-K (Bio-Rad) and evaluated on Personal Molecular Imager System (Bio-Rad). Representative radiograms of three experiments are displayed.

## Labeling and purification of substrate peptides

C4 peptide (RRYCKSTEL) was synthesized on solid phase using standard Fmoc chemistry. For fluorophore labeling, C4 peptide was incubated with a 1.2 molar excess of 5-iodoacetamide-fluorescein (Sigma-Aldrich) or ATTO655-maleimide (ATTO-TEC) in PBS DMF buffer (8.1 mM $Na_2HPO_4$ pH 6.5, 137 mM NaCl, 2.7 mM KCl, 1.8 mM $KH_2PO_4$, 20% [v/v] DMF) for 1 hr at 20°C. Peptides were purified by reversed-phase HPLC (Jasco; PerfectSil 300 ODS $C_{18}$), utilizing a linear acetonitrile gradient from 5% to 80% supplemented with 0.1% (v/v) TFA. Purified fluorescently labeled peptides were snap-frozen in liquid nitrogen and freeze-dried (Lyovac GT2, Heraeus).

## Data presentation and statistics

All measurements were performed in triplicates (n = 3). All diagrams were prepared in GraphPad Prism5, and mean values ± SD were presented. Statistical analysis was performed in GraphPad Prism5 applying two-tailed t-tests.

## Acknowledgements

We thank Dr. Rupert Abele, Dr. Simon Trowitzsch, Dr. Lukas Sušac, Dr. Christoph Thomas, Andrea Pott, Inga Nold, and all members of the Institute for Biochemistry for discussion and many helpful comments. RO gratefully acknowledges support from a JSPS Grants-in-Aid for Research Fellowship (P15333). This research was also supported by Japan Society for the Promotion of Science (JSPS) Grant-in-Aid for Specially Promoted Research (JP20H05618 to HS) and the Japan Agency for Medical Research and Development (AMED), Platform Project for Supporting Drug Discovery and Life Science Research, Basis for Supporting Innovative Drug Discovery and Life Science Research (JP20am0101090 to HS). The support by the German Research Foundation (CRC 807/P24 to NM; CRC 807/P16 to RT), the Reinhart Koselleck Project (TA 157/12–1 to RT), by LOEWE DynaMem A03, and by an ERC Advanced Grant (789121 to RT) of the European Research Council is gratefully acknowledged.

## Additional information

### Funding

| Funder | Grant reference number | Author |
| --- | --- | --- |
| Deutsche Forschungsgemeinschaft | TA 157/12-1 | Robert Tampé |
| Deutsche Forschungsgemeinschaft | CRC 807/P16 | Robert Tampé |
| European Research Council | 789121 | Robert Tampé |
| LOEWE DynaMem | A03 | Robert Tampé |
| Japan Society for the Promotion of Science | JP20H05618 | Hiroaki Suga |
| Japan Society for the Promotion of Science | JP20am0101090 | Hiroaki Suga |
| JSPS Grants-in-Aid for Research Fellowship | P15333 | Richard Obexer |
| Deutsche Forschungsgemeinschaft | CRC 807/P24 | Nina Morgner |

The funders had no role in study design, data collection and interpretation, or the decision to submit the work for publication.

### Author contributions

Erich Stefan, Resources, Data curation, Formal analysis, Validation, Investigation, Visualization, Methodology, Writing - original draft, Writing - review and editing, Performed all protein biochemistry experiments; Richard Obexer, Resources, Data curation, Formal analysis, Validation, Investigation, Visualization, Methodology, Writing - review and editing, Designed and carried out the random nonstandard peptide integrated discovery (RaPID) approach to identify and synthesize macrocyclic peptides; Susanne Hofmann, Resources, Data curation; Khanh Vu Huu, Data curation, Formal analysis, Performed the native MS analyses; Yichao Huang, Data curation, Synthesize macrocyclic peptides; Nina Morgner, Formal analysis, Supervision, Funding acquisition, Validation, Performed the native MS analyses; Hiroaki Suga, Conceptualization, Formal analysis, Supervision, Funding acquisition, Validation, Project administration, Writing - review and editing, Designed and carried out the random nonstandard peptide integrated discovery (RaPID) approach to identify and synthesize macrocyclic peptides; Robert Tampé, Conceptualization, Formal analysis, Supervision, Funding acquisition, Validation, Investigation, Visualization, Methodology, Writing - original draft, Project administration, Writing - review and editing, Initiated, supervised, and conceived the study

Author ORCIDs
Nina Morgner (iD) https://orcid.org/0000-0002-1872-490X
Robert Tampé (iD) https://orcid.org/0000-0002-0403-2160

Decision letter and Author response
Decision letter https://doi.org/10.7554/eLife.67732.sa1
Author response https://doi.org/10.7554/eLife.67732.sa2

## Additional files

### Supplementary files

• Transparent reporting form

### Data availability

All data denerated or analyzed during this study are included in the manuscript and support files. A source data file has been provided for Figure 1C (Sequencing Data), Figure 2-6, Figure 2-supplement figure 1 and Figure 5-supplement figure 1.

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
