## [Decision Letter]

**Acceptance summary:**

The RaPID technology is applied on the heterodimeric ABC transporter TmrAB from Thermus thermophilus to generate high-affinity macrocyclic peptides (MCPs) with conformational-specific preferences. The MCPs reveal novel, allosteric binding sites in TmrAB, which do not inhibit ATP or substrate binding, but block ATP hydrolysis. The MCPs are used as tool compounds to provide evidence that ATP binding, rather than ATP hydrolysis, is sufficient to drive substrate export in this highly-coupled ABC exporter. The authors further demonstrate that MCPs can be generated for membrane proteins embedded in nanodiscs, rather than in detergent, which is an important technical step for peptide-based drug discovery.

**Decision letter after peer review:**

Thank you for submitting your article "De-novo macrocyclic peptides dissect energy coupling of a heterodimeric ABC transporter by multimode allosteric inhibition" for consideration by *eLife*. Your article has been reviewed by 3 peer reviewers, including David Drew as the Reviewing Editor and Reviewer #1, and the evaluation has been overseen by Olga Boudker as the Senior Editor.

Essential revisions:

This is an excellent technical and mechanistic work describing the discovery and characterization of macrocyclic peptides (MCPs) that target the heterodimeric ABC transporter TmrAB from *Thermus thermophilus*. It is significant because it reports on the discovery of at least two distinct classes of MCPs that trap TmrAB in distinct states, and the authors use these unique MCPs as powerful probes to provide new insight into the catalytic mechanism of TmrAB. The MCPs reveal novel allosteric binding sites in TmrAB and provides evidence that ATP binding, rather than ATP hydrolysis, is sufficient to drive substrate export in this highly-coupled ABC exporter. Overall, this work 1) introduces potent and selective modulators of the model TmrAB transporter, 2) provides key mechanistic insight into the still unsettled transport mechanism of TmrAB through pharmacological arrest, and 3) highlights the potential of MCPs as pharmacological tools and their potential as early drug leads.

1. While Figure 5A/B are very welcome data, the authors do not systematically determine the affinity of all four MCPs in the liposome and detergent conditions that are used interchangeably throughout this study. Given the potential impact on Kd (i.e. CP6 shows an apparent 6-60-fold decrease in affinity from the liposomes to DDM condition), it seems of particular importance that the K_off_ for CP14 can be established (ideally at 45 degrees) in order to fully understand it's propensity to disassociate (i.e. bind/unbind) during the time course and temperature changes of the experiments presented in Figure 5C/D, and Figure 6E, since this has clear implications for the authors present interpretations as summarized in Figure 7. We would expect SPR experiments in detergent and liposomes (or nanodiscs) are important data to present "IF" these experiments are technically achievable. If not, Ki measurements of the CP's for C4-peptide uptake in liposomes with 50:50 orientation, should be able to provide a proxy for the binding affinity that should correlate with the preferences of the peptides for the OF or IF states as determined here in detergent.

Comments for authors:

Manuscript strengths:

1. The authors build on their prior work on TmrAB to present and utilize an arsenal of well-established biochemical and biophysical assays throughout the manuscript.

2. In general, the authors are thoughtful in their use of these novel MCPs to characterize the mechanism of action on TmrAB and to provide key and novel insights into the TmrAB transport mechanism.

3. The scope of discovery and characterization presented in this work is rather unique for any membrane transport system; it will undoubtedly be of high interest to the ABC transporter field and to membrane biologists in general.

Manuscript weaknesses; support of claims and conclusions:

1. While Figure 5A/B are very welcome data, the authors do not systematically determine the affinity of all four MCPs in the liposome and detergent conditions that are used interchangeably throughout this study. Given the potential impact on Kd (i.e. CP6 shows an apparent 6-60-fold decrease in affinity from the liposomes to the detergent condition), it seems of particular importance that the K_off_ for CP14 can be established (ideally at 45 degrees) in order to fully understand it's propensity to disassociate (i.e. bind/unbind) during the time course and temperature changes of the experiments presented in Figure 5C/D, and Figure 6E, since this has clear implications for the authors present interpretations as summarized in Figure 7.

2. Quantitative analysis (Figure 5): CP13 has an affinity of 40 nM for the IF and 10 nM for OF. Why is it selective for OF? Since the concentration of the CP13 used in the assay is above 40 nM does this not imply that the IF state will also be recognized and trapped at this concentration? Does this really mean CFs are conformation specific or would it be more correct to conclude that they have conformational-specific preferences?

3. Would different macrocyclic peptides have been generated if TmrAB was presented in detergent and the wild type protein was used instead? Whilst this is not a critical comparative control and the beneficial benefits of using nanodiscs are obvious, it would have helped to have included this control to fully benchmark the new methodological developments.

4. The currently described MCPs are not able membrane permeable and so cannot be used in vivo without further chemical modification.

5. As stated in line 129/130, all CPs analyzed blocked ATPase activity to the level of autohydrolysis. If the CPs abolished ATPase activity how is it possible that substrate transport is only partially inhibited by CP13 and CP14 (line 135/136) since transport depends on ATPase activity?

6. It is unclear why the CP13 has a coupling ratio of 0.3. Does it mean that 30% of the transporters are not inhibited by CP13? Or does CP13 imposes inefficient coupling?

7. Figure 3C and 3D: There are higher levels of transported peptide as compared to the WT protein when is ADP is added. It is unclear if the amount of transported CF4 in the presence of CPs really "background" as claimed by the authors?

---

## [Author Response]

Essential revisions:[…]1. While Figure 5A/B are very welcome data, the authors do not systematically determine the affinity of all four MCPs in the liposome and detergent conditions that are used interchangeably throughout this study. Given the potential impact on Kd (i.e. CP6 shows an apparent 6-60-fold decrease in affinity from the liposomes to DDM condition), it seems of particular importance that the K_off_ for CP14 can be established (ideally at 45 degrees) in order to fully understand it's propensity to disassociate (i.e. bind/unbind) during the time course and temperature changes of the experiments presented in Figure 5C/D, and Figure 6E, since this has clear implications for the authors present interpretations as summarized in Figure 7. We would expect SPR experiments in detergent and liposomes (or nanodiscs) are important data to present "IF" these experiments are technically achievable. If not, Ki measurements of the CP's for C4-peptide uptake in liposomes with 50:50 orientation, should be able to provide a proxy for the binding affinity that should correlate with the preferences of the peptides for the OF or IF states as determined here in detergent.

We would like to thank the reviewers for these insightful comments and the very positive feedback, highlighting the importance of the macrocyclic inhibitors, which can be used as potential pharmacological tools and also provide key mechanistic insights into heterodimeric ABC transporters.

To investigate the kinetic stability of detergent-solubilized TmrAB-CP complexes, we performed gel filtration, which routinely takes up to 45 min (Figure 2—figure supplement 1C). Over this period of time, we detected the co-elution of CPs and TmrAB with minute amounts of unbound CPs, demonstrating the formation of kinetically stable complexes. With regard to the experiments shown in Figure 5C/D, nucleotide occlusion was recorded immediately after trapping and rapid gel filtration. In Figure 6E, the proteoliposomes were not washed and the CPs were added at 1 µM, which was up to 50-fold above the equilibrium dissociation constant of CPs to yield high occupancy. In both cases (Figure 5C/D and 6E), CPs were added in excess, and no extensive washing steps were introduced.

Therefore, we assumed a high occupancy of CPs in the course of the experiments, as depicted in Figures 5C/D and 6E. It is technically extremely challenging and demanding to investigate the accurate binding and dissociation rates by SPR for small molecules and large membrane protein complexes. Although the determination of these rates might increase the impact of this work, we do not feel, that in the COVID-19 situation, this information would largely change the conclusion described in the initial submission. In addition, we have characterized two mechanistically similar inhibitors, additional experiments with the inferior inhibitor CP14 would not produce any other results.

Comments for authors:[…] Manuscript weaknesses; support of claims and conclusions:1. While Figure 5A/B are very welcome data, the authors do not systematically determine the affinity of all four MCPs in the liposome and detergent conditions that are used interchangeably throughout this study. Given the potential impact on Kd (i.e. CP6 shows an apparent 6-60-fold decrease in affinity from the liposomes to the detergent condition), it seems of particular importance that the K_off_ for CP14 can be established (ideally at 45 degrees) in order to fully understand it's propensity to disassociate (i.e. bind/unbind) during the time course and temperature changes of the experiments presented in Figure 5C/D, and Figure 6E, since this has clear implications for the authors present interpretations as summarized in Figure 7.

**Please see our comment above.**

2. Quantitative analysis (Figure 5): CP13 has an affinity of 40 nM for the IF and 10 nM for OF. Why is it selective for OF? Since the concentration of the CP13 used in the assay is above 40 nM does this not imply that the IF state will also be recognized and trapped at this concentration? Does this really mean CFs are conformation specific or would it be more correct to conclude that they have conformational-specific preferences?

The reviewer is right, and we already carefully rephrased the conformational-specific preferences in the main text of our original manuscript. In addition, we corrected the first sentence of the legend of Figure 5 and changed ‘conformation specific’ to ‘conformational-specific preference’ (line 760).

3. Would different macrocyclic peptides have been generated if TmrAB was presented in detergent and the wild type protein was used instead? Whilst this is not a critical comparative control and the beneficial benefits of using nanodiscs are obvious, it would have helped to have included this control to fully benchmark the new methodological developments.

We addressed this aspect in the detailed point-to-point reply.

4. The currently described MCPs are not able membrane permeable and so cannot be used in vivo without further chemical modification.

These peptides were only characterized in in vitro systems. In this context, passive membrane permeability can be excluded. Passive membrane permeability is dependent on peptide conformations that shield all groups involved in hydrogen bond formation (including backbone amides and polar side chains) so that the peptide can, in rare cases, partition into the lipid bilayer. With regard to in vivo contexts, cell permeability is also achieved via endocytosis or endocytosis-like pathways amongst other mechanisms. In this regard, cell permeability has been observed for various RaPID peptides (e.g. Kawamura *et al.* DOI: 10.1038/ncomms14773, Dai *et al.* DOI:10.1038/ncomms14773). Since we did not perform in vivo experiments, which are also beyond the scope of this work, we cannot comment on cell permeability.

5. As stated in line 129/130, all CPs analyzed blocked ATPase activity to the level of autohydrolysis. If the CPs abolished ATPase activity how is it possible that substrate transport is only partially inhibited by CP13 and CP14 (line 135/136) since transport depends on ATPase activity?

This question brings up a very important aspect of our manuscript, because many mechanistic determinants of asymmetric ABC transporters remain elusive: What defines the origin of the power stroke that drives unidirectional substrate transport? Which factors control the energy coupling of consumed ATP per translocated substrate? Which conformational transition drives unidirectional substrate transport? To address these key questions in the field, we evolved high-affinity, conformation-selective macrocyclic peptides by combinatorial design using the RaPID system and deep sequencing. The identified macrocyclic peptides arrest the ABC transport complex before and after the power stoke trajectory. For the first time to our knowledge, macrocyclic peptide inhibitors are used to realize single-turnover analyses of a wildtype ABC transporter to extend quantitative mechanistic determinants, a strategy which has not yet been realized with antibody fragments.

In our study, we demonstrate that selected macrocycles are not simple competitive inhibitors as they block ATP hydrolysis and substrate transport at nanomolar concentrations, while retaining ATP or substrate binding. Macrocycles interact in an allosteric, conformation-selective manner and stabilize a pre-hydrolysis state of the ABC transporter. By employing these macrocyclic peptides, the transport cycle can be dissected in single-turnover events, demonstrating an energetically coupled substrate transport step of a wildtype transporter along a single IF-to-OF switch induced by ATP binding.

Single-turnover studies are of immediate relevance to membrane transporters and highlight the potential of peptidic macrocycles as next-generation drugs targeting elaborate membrane protein complexes implicated in multidrug resistance. We outline a general framework to identify selective inhibitors against delicate membrane proteins that can be applied to almost any medically relevant protein complex.

6. It is unclear why the CP13 has a coupling ratio of 0.3. Does it mean that 30% of the transporters are not inhibited by CP13? Or does CP13 imposes inefficient coupling?

This is most likely inefficient coupling because, so far, many ABC export system have been described with very inefficient energetic coupling ratios.

7. Figure 3C and 3D: There are higher levels of transported peptide as compared to the WT protein when is ADP is added. It is unclear if the amount of transported CF4 in the presence of CPs really "background" as claimed by the authors?

Thank you for mentioning this potentially misleading point. We changed Figure 3C and included labels stating that peptide C4F was analyzed in ATP-dependent transport in the absence and presence of DMSO. In Figure 3C/D, the indicated transporter peptides of *0.25-0.5 µmol / g / min* are background but also potentially binding to TmrAB as slight differences between ADP and ATP can be observed, indirectly reflecting preferential binding to the IF and OF conformation of TmrAB. We also rephrase the legend of Figure 3C.